# INTEGRATIVE DECODING: IMPROVE FACTUALITY VIA IMPLICIT SELF-CONSISTENCY

**Yi Cheng**[1]***Xiao Liang**[2], **Yeyun Gong**[3], **Wen Xiao**[4], **Song Wang**[4], **Yuji Zhang**[5], **Wenjun Hou**[1],
**Kaishuai Xu**[1], **Wenge Liu**[1], **Wenjie Li**[1], **Jian Jiao**[3], **Qi Chen**[3], **Peng Cheng**[3], **Wayne Xiong**[3]
[1]The Hong Kong Polytechnic University   [2]Tsinghua University   [3]Microsoft Research
[4]Microsoft Azure AI   [5]University of Illinois at Urbana-Champaign
`alyssa.cheng@connect.polyu.hk`

## ABSTRACT

Self-consistency-based approaches, which involve repeatedly sampling multiple outputs and selecting the most consistent one as the final response, prove to be remarkably effective in improving the factual accuracy of large language models. Nonetheless, existing methods usually have strict constraints on the task format, largely limiting their applicability. In this paper, we present *Integrative Decoding* (ID), to unlock the potential of self-consistency in open-ended generation tasks. ID operates by constructing a set of inputs, each prepended with a previously sampled response, and then processes them concurrently, with the next token being selected by aggregating of all their corresponding predictions at each decoding step. In essence, this simple approach implicitly incorporates self-consistency in the decoding objective. Extensive evaluation shows that ID consistently enhances factuality over a wide range of language models, with substantial improvements on the TruthfulQA (+11.2%), Biographies (+15.4%) and LongFact (+8.5%) benchmarks. The performance gains amplify progressively as the number of sampled responses increases, indicating the potential of ID to scale up with repeated sampling.[1]

## 1 INTRODUCTION

Despite notable advancements across various domains, Large Language Models (LLMs) remain notorious for their tendency to produce non-factual and erroneous content, a phenomenon commonly known as hallucinations (Lewis et al., 2020; Ji et al., 2023). Prior research has shown that "repeated sampling" is a very effective methodology for enhancing factual accuracy (Wang et al., 2023; Shi et al., 2022; Chen et al., 2023). It involves sampling multiple responses to the same prompt, followed by a careful selection of the most accurate one or the synthesis of a refined output from the sampled responses. Notably, as the number of sampled responses increases, its performance gains often continue to rise in an almost log-linear manner, as recently highlighted by Brown et al. (2024). This suggests the existence of "inference-time scaling laws," implying the potential of repeated sampling to progressively push the model closer to its theoretical performance ceilings. Despite this immense promise, a central challenge in this methodology remains: how to effectively identify the non-factual content within the sample collection and thereby produce a final, accurate output.

The degree of "*self-consistency*" (SC), which measures the consistency level among LLMs' different outputs, has proven to be a useful indicator to address this issue (Wang et al., 2023; Shi et al., 2022; Chen et al., 2023; Thirukovalluru et al., 2024; Malon & Zhu, 2024; Mündler et al., 2024; Manakul et al., 2023). It has been observed that statements consistently present across a range of sampled responses are more likely to be truthful, as opposed to those appearing sporadically or inconsistently across outputs. However, most SC-based methods for improving factuality impose strict constraints on the format of task output, largely limiting their applicability. Due to the difficulty in measuring consistency across responses, previous studies usually only consider tasks where they can easily define consistency as the exact matches between the answers parsed from the responses (Wang et al., 2023; Huang et al., 2023a; Shi et al., 2022; Li et al., 2022), such as arithmetic problems and

---

*This work was conducted during Yi Cheng's internship at Microsoft Research.
[1]All codes and data are available at https://github.com/YiCheng98/IntegrativeDecoding.

Table 1: Comparisons between ID and previous approaches that utilize self-consistency to improving factuality on open-ended-generation tasks. "Input length" indicates the length relative to that of one sampled response from standard prompting (with $k$ representing the number of sampled responses).

| Method | How to Check Self-consistency | Input Length | Inference Latency | Balance Informativeness | Factuality Improvement |
|---|---|---|---|---|---|
| USC (Chen et al., 2023) | Prompting | $\times k$ | Medium | ✓ | Medium |
| SR (Madaan et al., 2024) | CoT Reasoning | $\times k$ | Medium | ✗ | Medium |
| FSC (Wang et al., 2024a) | CoT Reasoning | $\times k$ | Medium | ✗ | High |
| SE-SL (Wang et al., 2024b) | Numerous Prompting | $\times 1$ | High | ✓ | High |
| SE-RG (Wang et al., 2024b) | Prompting & Clustering | $\times 1$ | High | ✗ | High |
| Integrative Decoding | ICL & Decoding-time Implicit Integration | $\times 1$ | Medium | ✓ | Higher |

multiple choice question. This naturally leads us to ask: *how can we further unlock the potential of self-consistency and repeated sampling in open-ended generation tasks*?

One straightforward way is to concatenate all sampled responses in a prompt and directly instruct the LLM to select the most self-consistent one from them, as done in Chen et al. (2023). Nonetheless, such practice substantially increases the input length, posing excessive demands on the model's long-text processing capability. Another line of research treats each response as a collection of statements and then assess the consistency level between each pair of statements through clustering (Thirukovalluru et al., 2024) or iterative LLM prompting (Mündler et al., 2024; Wang et al., 2024a;b). This requires numerous iterations of inference, particularly for longer outputs, leading to inefficiencies. Due to these issues, prior attempts to apply SC in open-ended tasks cannot generalize effectively to long-form generations and they struggle to scale up with an increasing number of sampled responses.

In this paper, we present *Integrative Decoding* (ID), a novel decoding strategy designed to improve factuality by implicitly incorporating self-consistency within its decoding objective. ID begins by repeated sampling. For each sampled response in the collection, ID constructs a new input by concatenating the response with the original prompt. Essentially, this input instructs the model to respond to the instruction again with reference to a previously sampled response. Then, ID processes these inputs concurrently for decoding, with the next token being selected by integrating all their predictions at each inference step. During this process, each input acts like a "representative" for the sampled response within it, voting for the tokens that are semantically consistent with the response it represents. ID effectively aggregates their votes and thereby achieves the optimal overall consistency across all sampled responses. Compared with existing approaches that utilize self-consistency to improve factuality on open-ended generation tasks, ID does not rely on additional prompting or chain-of-thought reasoning to explicitly verify consistency; moreover, it can achieve substantial improvement in factuality with relatively low inference latency and a slight burden on the model's long-text processing capabilities (see Table 1 for detailed comparisons).

We evaluate ID over six series of LLMs with varying scales. ID consistently enhances the factuality over all these LLMs by a large margin on the TruthfulQA (+11.2%), Biographies (+15.4%) and LongFact (+8.5%) datasets, demonstrating robustness from sentence- to document-level generations. Moreover, the performance gains of ID progressively amplify as the number of sampled responses increases, indicating its potential to scale up with repeated sampling.

## 2 METHOD

**Preliminaries: Self-consistency as an Indicator for Factuality** Previous studies found that the degree of self-consistency between LLM's different sampled responses can serve as a useful indicator for hallucination detection (Manakul et al., 2023; Farquhar et al., 2024). The facts that are consistently supported by LLMs' different sampled responses are more likely to be factual, compared to those that only appear sporadically or inconsistently across multiple outputs. Formally, given a prompt $\mathbf{x}$ and its response $\hat{\mathbf{y}}$ that consists of a series of statements $\mathcal{S} = \{s_1, s_2, .., s_n\}$, the factuality score of $s_i$ can be estimated by measuring its consistency with other sampled responses

$\mathcal{R} = \{r_1, r_2, .., r_k\}$ in response to the same prompt $\mathbf{x}$ as:

$$f(s_i) = \frac{1}{|\mathcal{R}|} \sum_{r_j \in R} P(\text{consistent}|s_i, r_j), \tag{1}$$

where $f(s_i)$ refers to the estimated factuality score of the statement $s_i$ and $P(\text{consistent}|s_i, r_j)$ is the probability that $s_i$ is supported by the response $r_j$. These responses can be obtained through sampling algorithms, such as temperature sampling (Ficler & Goldberg, 2017) or nucleus sampling Holtzman et al. (2020). The overall factuality score of the response $\hat{\mathbf{y}}$ can thereby be estimated as:

$$F(\hat{\mathbf{y}}) = \frac{1}{|\mathcal{S}| \cdot |\mathcal{R}|} \sum_{s_i \in \mathcal{S}} \sum_{r_j \in R} P(\text{consistent}|s_i, r_j) = \frac{1}{|\mathcal{R}|} \sum_{r_j \in \mathcal{R}} \bar{f}(\hat{\mathbf{y}}, r_j), \tag{2}$$

where $\bar{f}(\hat{\mathbf{y}}, r_j) = \frac{1}{|\mathcal{S}|} \sum_{s_i \in \mathcal{S}} P(\text{consistent}|s_i, r_j)$, representing the overall degree of $\hat{\mathbf{y}}$ being supported by the response $r_j$.

**Formalization of Decoding Objective**   The established insights about the role of self-consistency in hallucination detection indicate that the response most consistent with the others tends to be the most factual one. This motivates us to develop a decoding method that, given several sampled responses, can generate a new output, maintaining strong overall consistency with all of them while maintaining its own coherence. Formally, given an input prompt $\mathbf{x}$, a decoding method searches for an output $\hat{\mathbf{y}}$ by solving:

$$\hat{\mathbf{y}} = \arg\max_{\mathbf{y} \in \mathcal{Y}} H(\mathbf{x}, \mathbf{y}), \tag{3}$$

where $\mathcal{Y}$ refers to the set of all possible token sequences and $H(\mathbf{x}, \mathbf{y})$ is the objective function.

Common decoding algorithms, such as beam search, consider the decoding objective $H(\mathbf{x}, \mathbf{y})$ as $\log p_\theta(\mathbf{y}|\mathbf{x}) = \sum_{t=1}^{|\mathbf{y}|} \log p_\theta(y_t|y_{<t}, \mathbf{x})$, where $\theta$ refers to the model's parameters and $p_\theta(y_t|y_{<t}, \mathbf{x})$ represents its predicted token probability distribution at the $t$-th decoding step. Note that we omit the input prompt $\mathbf{x}$ here and in the following to reduce clutter.

The objective of our method, by contrast, is composed of two parts: $H(\mathbf{x}, \mathbf{y}) = F(\mathbf{y}) + \lambda \cdot G(\mathbf{x}, \mathbf{y})$, where $\lambda$ is a constant weight. $G(\mathbf{x}, \mathbf{y})$ can be viewed as the common decoding objective, which measures whether the concatenation of $\mathbf{x}$ and $\mathbf{y}$ is a coherent and contextually appropriate text. $F(\mathbf{y})$ is used to measure truthfulness of $\hat{\mathbf{y}}$, which additionally emphasizes factuality in the decoding objective. Then, we adapt this objective function by replacing $F(\mathbf{y})$ based on Equation 2:

$$H(\mathbf{y}) = \sum_{r_j \in R} [\bar{f}(\mathbf{y}, r_j) + \alpha \cdot G(\mathbf{x}, \mathbf{y})], \tag{4}$$

where $\mathcal{R}$ is a set of sampled responses to the prompt $\mathbf{x}$ and $\alpha$ is a constant term.

**Integrative Decoding**   However, computing Equation 4 directly poses significant challenges, especially for the part of $\bar{f}(\mathbf{y}, r_j)$. Previous studies typically rely on LLMs to ascertain whether the statements in $\mathbf{y}$ are supported by $r_j$ (Mündler et al., 2024; Manakul et al., 2023). This process is not only computationally expensive, but also requires sophisticated prompt design to comprehensively measure $\bar{f}(\mathbf{y}, r_j)$.

To address this, our method incorporates an estimation of Equation 4 as follows. Crucially, the part of $\bar{f}(\hat{\mathbf{y}}, r_j) + \alpha \cdot G(\mathbf{x}, \mathbf{y})$ in Equation 4 is approximated as the LLM's predicted probability for the output sequence when instructed to *respond to* $\mathbf{x}$ *again with reference to a previously sampled response* $r_j$. Specifically, this involves constructing a new input $q_j$, which is sequentially structured as $[\mathbf{x}; r_j; \mathbf{x}]$.[2] Formally, we assume that:

$$\log p_\theta(\mathbf{y}|[\mathbf{x}; r_j; \mathbf{x}]) \propto \bar{f}(\mathbf{y}, r_j) + \alpha \cdot G(\mathbf{x}, \mathbf{y}). \tag{5}$$

This assumption is reasonable because when $q_j$ serves as the input, the LLM's in-context learning abilities naturally incline it to produce content consistent with $r_j$ within the input, thus promoting

---

[2] Note that, in practice, $q_j$ is not a strict concatenation of $\mathbf{x}$, $r_j$, and $\mathbf{x}$. Additional clarifying instructions, such as "answer this question again", need to be inserted after $r_j$ to avoid confusion. We omit these details in the representation of $q_j$ here to reduce clutter.

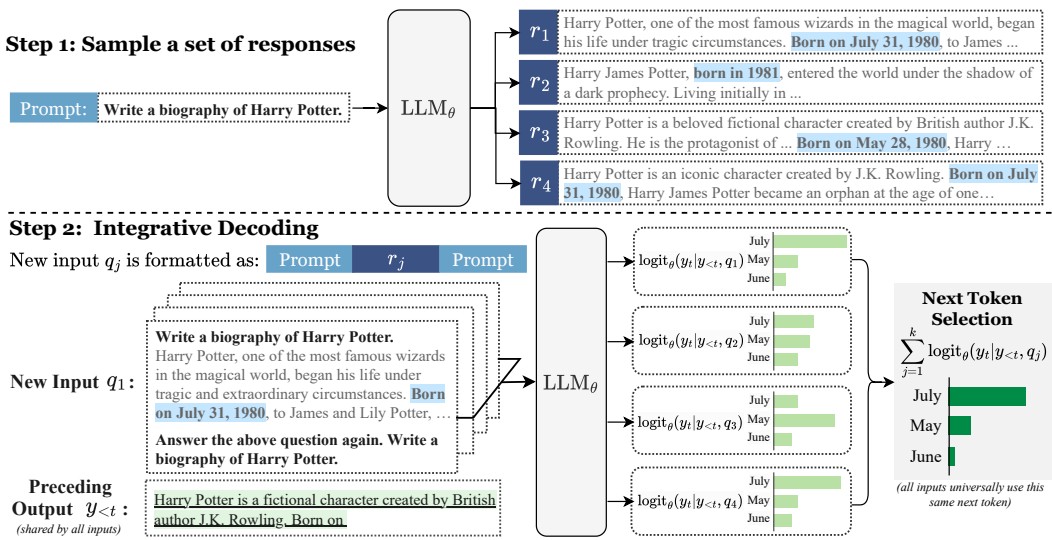

Figure 1: The workflow of integrative decoding: (1) sample multiple responses from the LLM; (2) form a set of new inputs by concatenating a sampled response and the original prompt; they are concurrently processed for decoding, with the next token being selected by integrating their predicted logits at each inference step. This strategy essentially incorporates the overall consistency with all sampled responses in its decoding objective (see Section 2).

$\bar{f}(\mathbf{y}, r_j)$. Concurrently, the LLM also ensures that the combination $\mathbf{x} \circ \mathbf{y}$ remains coherent and contextually appropriate, enhancing $G(\mathbf{x}, \mathbf{y})$. In other words, the LLM tends to choose the output that is not only consistent with $r_j$ but also maintains its own coherence. This supports the validity of Equation 5 as a plausible assumption.

Then, we replace Equation 4 with:

$$H(y) = \sum_{r_j \in R} \log p_\theta(\mathbf{y}|[\mathbf{x}; r_j; \mathbf{x}]). \tag{6}$$

which ideally should be computed as:

$$H(\mathbf{y}) = \sum_{r_j \in R} \sum_{t=1}^{|\mathbf{y}|} \log p_\theta(y_t|y_{<t}, [\mathbf{x}; r_j; \mathbf{x}]), \tag{7}$$

Nonetheless, due to the prohibitively large searching space for $y \in \mathcal{Y}$, it is extremely difficult to compute Equation 7. To enhance computational efficiency, we adopt the strategy commonly used in greedy algorithms by making locally optimal decisions at each decoding step. Specifically, at the $t$-th decoding step, we choose the next token $\hat{y}_t$ by:

$$\hat{y}_t = \arg\max_{y_t \in \mathcal{V}} \sum_{r_j \in R} \log p_\theta(y_t|y_{<t}, [\mathbf{x}; r_j; \mathbf{x}]). \tag{8}$$

Based on the above analysis, we can summarize the workflow to produce the result $\hat{\mathbf{y}}$ as dipicted in Figure 1. It begins by sampling multiple responses $\mathcal{R} = \{r_1, r_2, .., r_k\}$ and then constructing a set of new inputs $\mathcal{Q} = \{q_1, q_2, ..., q_k\}$ to prompt the model respond to the orginal instruction again with reference to a previously sampled response. Subsequently, these inputs are fed to the LLM, which can be processed in one batch concurrently. At the $t$-th decoding step, we integrate all predicted probability logits in this batch and select the next token as illustrated in Equation 8. All sequences in the batch universally take the same next token and then continue the generation process. Consequently, all inputs in the batch result in the same output $\hat{\mathbf{y}}$, which is used as the final response to the prompt $\mathbf{x}$.

Table 2: Evaluation results on three open-ended benchmarks. Responses on TruthfulQA are brief *sentences*, Biographies are short *paragraphs*, and LongFact requires *document-level* responses. The three benchmarks pose increasing levels of difficulty for factuality enhancement. The best results are highlighted in blue and the second best are in green. The results indicating a performance drop (i.e., worse than the standard greedy decoding) are marked in grey.

| Method | | TruthfulQA | | | Biographies | | LongFact | | |
|---|---|---|---|---|---|---|---|---|---|
| | | % Truth | % Info | % T*I | # Correct | % Acc. | Prec. | R@128 | F1@128 |
| LLaMA2 | Greedy | 50.7 | 96.3 | 48.9 | 0.81 | 16.2 | 88.1 | 75.6 | 80.5 |
| | DoLA | 49.5 (-1.2) | 95.6 (-0.7) | 47.3 (-1.6) | 0.78 (-0.03) | 15.6 (-0.6) | 88.0 (-0.1) | 75.5 (-0.1) | 80.4 (-0.1) |
| | USC | 46.3 (-4.4) | 96.1 (-0.2) | 44.5 (-4.4) | 0.84 (+0.03) | 16.7 (+0.5) | 86.5 (-1.6) | 72.1 (-3.5) | 77.6 (-2.9) |
| | SR | 53.9 (+3.2) | 96.3 (+0.0) | 51.9 (+3.0) | 0.82 (+0.01) | 16.6 (+0.4) | 86.8 (-1.3) | 58.2 (-17.4) | 55.0 (-25.5) |
| | SE-SL | 50.5 (-0.2) | 96.1 (-0.2) | 48.5 (-0.4) | 0.75 (-0.06) | 15.0 (-1.2) | 88.2 (+0.1) | 74.7 (-0.9) | 81.1 (+0.6) |
| | SE-RG | 45.4 (-5.3) | 94.6 (-1.7) | 42.9 (-6.0) | 0.82 (+0.01) | 16.4 (+0.2) | 85.2 (-2.9) | 54.5 (-21.1) | 64.8 (-15.7) |
| | FSC | 52.4 (+1.7) | 95.6 (-0.7) | 50.1 (+1.2) | 0.82 (+0.01) | 16.4 (+0.2) | 88.0 (-0.1) | 64.0 (-11.6) | 72.6 (-7.9) |
| | ID | 55.9 (+5.2) | 99.0 (+2.7) | 55.3 (+6.4) | 0.87 (+0.06) | 17.3 (+1.1) | 89.0 (+0.9) | 77.5 (+1.9) | 82.1 (+1.6) |
| LLaMA3 | Greedy | 53.4 | 96.6 | 51.6 | 1.28 | 26.6 | 90.0 | 70.7 | 78.8 |
| | DoLA | 54.1 (+0.7) | 97.6 (+1.0) | 52.8 (+1.2) | 1.30 (+0.02) | 27.1 (+0.5) | 90.3 (+0.3) | 70.5 (-0.2) | 78.8 (+0.0) |
| | USC | 56.8 (+3.4) | 98.3 (+1.7) | 55.9 (+4.3) | 1.34 (+0.06) | 27.9 (+1.3) | 89.7 (-0.3) | 71.8 (+1.1) | 79.3 (+0.5) |
| | SR | 57.8 (+4.4) | 97.1 (+0.5) | 56.1 (+4.5) | 1.62 (+0.34) | 34.0 (+7.4) | 89.4 (-0.6) | 46.1 (-24.6) | 58.6 (-20.2) |
| | SE-SL | 58.0 (+4.6) | 98.3 (+1.7) | 57.1 (+5.5) | 1.48 (+0.20) | 32.8 (+6.2) | 92.5 (+2.5) | 68.0 (-2.7) | 77.7 (-1.1) |
| | SE-RG | 54.4 (+1.0) | 96.3 (-0.3) | 52.4 (+0.8) | 1.60 (+0.32) | 34.5 (+7.9) | 91.8 (+1.8) | 47.7 (-23.0) | 62.0 (-16.8) |
| | FSC | 56.5 (+3.1) | 93.4 (-3.2) | 52.8 (+1.2) | 1.33 (+0.05) | 27.9 (+1.3) | 92.5 (+2.5) | 47.3 (-23.4) | 60.2 (-18.6) |
| | ID | 63.4 (+10.0) | 99.0 (+2.4) | 62.8 (+11.2) | 2.00 (+0.72) | 42.0 (+15.4) | 92.2 (+2.2) | 77.7 (+7.0) | 83.6 (+4.8) |
| Mistral2 | Greedy | 74.9 | 99.8 | 74.7 | 0.93 | 18.6 | 91.2 | 61.1 | 72.2 |
| | DoLA | 74.4 (-0.5) | 99.8 (+0.0) | 74.2 (-0.5) | 0.94 (+0.01) | 18.8 (+0.2) | 91.2 (+0.0) | 61.0 (-0.1) | 72.1 (-0.1) |
| | USC | 76.6 (+1.7) | 99.8 (+0.0) | 76.4 (+1.7) | 0.94 (+0.01) | 18.8 (+0.2) | 90.6 (-0.6) | 61.3 (+0.2) | 72.3 (+0.1) |
| | SR | 78.0 (+3.1) | 99.5 (-0.3) | 77.7 (+3.0) | 0.97 (+0.04) | 19.8 (+1.2) | 91.2 (+0.0) | 63.0 (+1.9) | 73.0 (+0.8) |
| | SE-SL | 76.8 (+1.9) | 99.5 (-0.3) | 76.8 (+2.1) | 1.16 (+0.23) | 23.3 (+4.7) | 91.6 (+0.4) | 58.5 (-2.6) | 70.6 (-1.6) |
| | SE-RG | 72.9 (-2.0) | 97.8 (-2.0) | 71.3 (-3.4) | 1.10 (+0.17) | 22.0 (+3.4) | 90.9 (-0.3) | 44.2 (-16.9) | 58.6 (-13.6) |
| | FSC | 78.0 (+3.1) | 99.5 (-0.3) | 77.7 (+3.0) | 0.87 (-0.06) | 17.5 (-1.1) | 91.3 (+0.1) | 57.8 (-3.3) | 69.1 (-3.1) |
| | ID | 78.8 (+3.9) | 99.5 (-0.3) | 78.4 (+3.7) | 1.11 (+0.18) | 22.6 (+4.0) | 91.8 (+0.6) | 68.5 (+7.4) | 77.7 (+5.5) |
| Qwen2 | Greedy | 56.3 | 97.1 | 54.7 | 1.45 | 29.1 | 90.0 | 57.1 | 69.1 |
| | DoLA | 56.1 (-0.2) | 96.6 (-0.5) | 54.2 (-0.5) | 1.46 (+0.01) | 29.2 (+0.1) | 89.5 (-0.5) | 56.6 (-0.5) | 68.7 (-0.4) |
| | USC | 58.3 (+2.0) | 97.6 (+0.5) | 56.9 (+2.2) | 1.44 (-0.01) | 28.8 (-0.3) | 87.9 (-2.1) | 57.3 (+0.2) | 68.7 (-0.4) |
| | SR | 59.8 (+3.5) | 97.6 (+0.5) | 58.3 (+3.6) | 1.42 (-0.03) | 28.6 (-0.5) | 85.0 (-5.0) | 45.8 (-11.3) | 57.5 (-11.6) |
| | SE-SL | 57.1 (+0.8) | 97.1 (+0.0) | 55.4 (+0.7) | 1.48 (+0.03) | 29.5 (+0.4) | 91.2 (+1.2) | 55.9 (-1.2) | 68.2 (-0.9) |
| | SE-RG | 62.9 (+6.6) | 94.9 (-2.2) | 59.7 (+5.0) | 1.54 (+0.09) | 30.8 (+1.7) | 91.3 (+1.3) | 44.3 (-12.8) | 57.9 (-11.2) |
| | FSC | 57.3 (+1.0) | 98.0 (+0.9) | 56.2 (+1.5) | 1.55 (+0.10) | 31.1 (+2.0) | 91.3 (+1.3) | 38.6 (-18.5) | 52.0 (-17.1) |
| | ID | 60.0 (+3.7) | 99.0 (+1.9) | 59.4 (+4.7) | 1.74 (+0.29) | 35.5 (+6.4) | 91.7 (+1.7) | 64.2 (+7.1) | 74.8 (+5.7) |
| Gemma2 | Greedy | 68.1 | 98.5 | 67.1 | 1.80 | 37.2 | 95.7 | 58.3 | 71.9 |
| | DoLA | 68.1 (+0.0) | 98.8 (+0.3) | 67.2 (+0.1) | 1.74 (-0.06) | 35.9 (-1.3) | 96.1 (+0.4) | 59.0 (+0.7) | 72.5 (+0.6) |
| | USC | 71.0 (+2.9) | 98.5 (+0.0) | 69.9 (+2.8) | 2.08 (+0.28) | 42.2 (+5.0) | 95.6 (-0.1) | 58.7 (+0.4) | 72.1 (+0.2) |
| | SR | 64.2 (-3.9) | 98.8 (+0.3) | 63.4 (-3.7) | 1.80 (+0.00) | 38.9 (+1.7) | 96.0 (+0.3) | 42.2 (-16.1) | 57.3 (-14.6) |
| | SE-SL | 69.8 (+1.7) | 98.3 (-0.2) | 68.3 (+1.2) | 2.29 (+0.49) | 47.3 (+10.1) | 97.1 (+1.4) | 56.1 (-2.2) | 70.3 (-1.6) |
| | SE-RG | 70.5 (+2.4) | 97.8 (-0.7) | 68.9 (+1.8) | 2.40 (+0.60) | 50.5 (+13.3) | 96.7 (+1.0) | 42.6 (-15.7) | 58.4 (-13.5) |
| | FSC | 69.8 (+1.7) | 98.3 (-0.2) | 68.3 (+1.2) | 1.70 (-0.10) | 36.0 (-1.2) | 95.8 (+0.1) | 50.4 (-7.9) | 65.1 (-6.8) |
| | ID | 77.1 (+9.0) | 99.0 (+0.5) | 76.3 (+9.2) | 2.52 (+0.72) | 52.4 (+15.2) | 97.1 (+1.4) | 69.7 (+11.4) | 80.4 (+8.5) |
| GLM4 | Greedy | 58.5 | 97.8 | 57.2 | 1.44 | 28.7 | 87.2 | 62.7 | 72.5 |
| | DoLA | 59.0 (+0.5) | 97.6 (-0.2) | 57.6 (+0.4) | 1.41 (-0.03) | 28.3 (-0.4) | 86.9 (-0.3) | 61.6 (-1.1) | 71.7 (-0.8) |
| | USC | 61.5 (+3.0) | 99.0 (+1.2) | 60.9 (+3.7) | 1.40 (-0.04) | 28.0 (-0.7) | 85.9 (-1.3) | 65.9 (+3.2) | 74.2 (+1.7) |
| | SR | 63.4 (+4.9) | 98.1 (+0.3) | 62.2 (+5.0) | 1.34 (-0.10) | 27.5 (-1.2) | 88.7 (+1.5) | 36.8 (-25.9) | 49.9 (-22.6) |
| | SE-SL | 61.0 (+2.5) | 98.5 (+0.7) | 60.1 (+2.9) | 1.37 (-0.07) | 27.3 (-1.4) | 88.9 (+1.7) | 62.5 (-0.2) | 72.9 (+0.4) |
| | SE-RG | 64.1 (+5.6) | 97.8 (+0.0) | 62.7 (+5.5) | 1.36 (-0.08) | 27.2 (-1.5) | 88.0 (+0.8) | 48.7 (-14.0) | 62.1 (-10.4) |
| | FSC | 63.4 (+4.9) | 97.8 (+0.0) | 62.0 (+4.8) | 1.58 (+0.14) | 31.7 (+3.0) | 90.3 (+3.1) | 38.4 (-24.3) | 52.8 (-19.7) |
| | ID | 65.1 (+6.6) | 99.0 (+1.2) | 64.5 (+7.3) | 1.81 (+0.37) | 36.2 (+7.5) | 89.2 (+2.0) | 66.4 (+3.7) | 75.9 (+3.4) |

## 3    EXPERIMENTS

### 3.1    SETUP

**Benchmarks and Evaluation Metrics**    We consider three open-ended generation benchmarks:
• **TruthfulQA** (Lin et al., 2022) consists of 817 questions that many humans would answer falsely due to misconception. We employ GPT-4 (Bubeck et al., 2023) to assess the truthfulness (*Truth*) and informativeness (*Info*) scores of each generated answer. The product of these two scores (*T*I*)

is considered as the major metric on this benchmark. During evaluation, the reference answers annotated in the dataset are included in the prompt as reference when using GPT-4 to assess truthfulness. The informativeness score assesses whether the response contains valid information that directly answers the question. GPT-4 is employed to evaluate this in a few-shot manner, using the evaluation samples provided by Lin et al. (2022) as the demonstration examples.

- **Biographies** (Du et al., 2024) requires generating bullet point biographies for computer scientists, with a total of 250 samples. Specifically, we prompt the model to list 5 major achievements or contributions made by the scientist in question. Following Du et al. (2024), we use GPT-4 to assess the factuality of each bullet statement by referring to the related information extracted from Wikipedia. The proportion (*%Accuracy*) and the number (*#Correct*) of factual statements are adopted as the evaluation metrics. Note that %Accuracy is not simply #Correct divided by five since the model may occasionally generate fewer than five statements when it is uncertain.

- **LongFact-Objects** (Wei et al., 2024) requests detailed descriptions for a queried object and expects a document-level response that is typically very long, often exceeding a thousand tokens (see Appendix G for detailed examples). The evaluation process is similar to the one described in Wei et al. (2024), which involves splitting the long response into a series of atomic facts and then assessing their truthfulness separately. We employ LLaMA3.1-70B-Instruct to divide atomic facts and use GPT-4 to assess whether each fact is truthful. The adopted metrics include the proportion of truthful facts (*Precision*), the number of truthful facts divided by 128 (*Recall@128*), and the *F1@128* score that integrates the previous two metrics. 120 samples are used for evaluation. Evaluation results of recall and F1 metrics at other intervals are provided in Appendix C.3.

**Compared Methods** We compare our method with (1) *greedy decoding* (**Greedy**) and (2) *decoding by contrasting layers* (Chuang et al., 2024b, **DoLa**). In addtion, we also compare it with five ensemble-based methods that also involves repeated sampling to produce a refined result, including: (3) *Universal Self-Consistency* (Chen et al., 2023, **USC**) concatenates the sampled responses in one prompt and directly instructs the LLM to select the most consistent one from them; (4) *Self-reflection* (Madaan et al., 2024, **SR**) also concatenates the sampled responses as an input, and asks the model to reflect on them and extract the factual information in them to produce a new response; (5) *Selection-based self-endorsement* (Wang et al., 2024b, **SE-SL**) prompts the LLM to divide the response into a sequence of facts and then calculates a self-endorsement score for each response by checking the consistency between each fact within it and all other sampled responses, selecting the response with the highest score as the final output; (6) *Regeneration-based self-endorsement* (**SE-RG**) is a variant of SE-SL, which regenerates a new output with some of the facts extracted from the sampled responses (7) *Fine-grained Self-consistency* (Wang et al., 2024a, **FSC**) instructs the LLM to extract common segments among sampled responses and regenerate a new output accordingly.

**Base Models** Our main experiments are conducted on LLaMA-2-7B-chat (Touvron et al., 2023), LLaMA-3-8B-Instruct (Dubey et al., 2024), Mistral-7B-Instruct-v0.2 (Jiang et al., 2023), Gemma-2-9B-it (Team et al., 2024), Qwen2-7B-Instruct (Yang et al., 2024), and GLM-4-9B-chat (GLM et al., 2024). We referto them as LLaMA2, LLaMA3, Mistral2, Gemma2, Qwen2, GLM4, respectively.

**Implementation Details** The prompt templates used for different approaches are provided in Appendix F. The sampled responses were all obtained via temperature sampling with $T = 0.7$ when implementing USC, SR, and ID in the main experiments. For USC, SR, and ID, we searched for the optimal number of sampled responses to integrate from $k = \{1, 4, 8, 12, 16\}$ using the validation sets and employ it for evaluation on the test sets. We selected the optimal $k$ according to the %Truth score on TruthfulQA and the %Accuracy metric on Biographies. Due to high evaluation costs on LongFact, we did not conduct optimal $k$ searching on it. We directly set $k = 16$ for ID. For USC, FSC and SR, we set $k = 4$ because these methods require including all sampled responses in the prompt. Since the responses on LongFact is very lengthy, setting $k$ higher than 4 would exceed the context length limits of many LLMs.

## 3.2 MAIN RESULTS

**Integrative decoding leads to substantial improvements in factuality across all six LLMs.** As shown in Table 2, the absolute improvements on TruthfulQA, Biographies, and LongFact are 3.7-10%, 1.1-15.4%, and 1.6-8.5%, respectively (in terms of %Truth, %Accuracy, and F1@128).

Among the six LLMs, the overall improvement is the most substantial over LLaMA3 and Gemma2. The improvement on LLaMA2, though evident, is the least among all six LLMs. This suggests that the effects of integrative decoding is more evident on stronger LLMs.

**Integrative decoding achieves robust balance between factuality and informativeness**. Across metrics that assess informativeness (i.e., %Info, #Correct, and Recall@128), integrative decoding also shows substantial improvement. This is particularly evident on the LongFact benchmark, which involves generating long documents, where the absolute improvement in Recall@128 reaches as high as 11.4%. This indicates that integrative decoding can elicit more parametric knowledge from the LLM while maintaining factual accuracy, rather than merely improving factuality simply by filtering out incorrect information. In contrast, the baseline methods, especially the other regeneration-based approaches (i.e., SR, FSC, SE-RG), struggle to achieve a robust balance between factuality and informativeness. For instance, while SR also improves the precision of GLM4 on LongFact, it results in a considerable drop of 25.9% in Recall@128. This indicates that they need to sacrifice a large degree of informativeness to ensure factual accuracy.

**Integrative decoding is robust to document-level generation tasks**. Enhancing factuality on long-form generation tasks is challenging and less explored. From Table 2, we can see that baseline approaches struggle with the LongFact benchmark, which requires document-level generation. Though some of them can also enhance precision, they often result in a marked decline in information recall the F1 metric. Encouragingly, integrative decoding remains effective on LongFact, providing absolute improvements of up to 8.5%. This suggests that integrative decoding offers greater generality and robustness in long-form generation tasks.

**Integrative decoding achieve more substantial and consistent improvement in factuality compared to the baseline approaches**. The improvements achieved by DoLa is marginal on our experimental benchmarks, with an increase of no more than 0.7%. This suggest that the effectiveness of DoLa in enhancing factuality is limited in long-form, open-ended generation tasks. While the other approaches can improve factual accuracy in many cases, their enhancements are not robust. They fail to reliably enhance performance across different LLMs; for instance, USC causes significant performance degradation on LLaMA2, and SR does the same on Gemma2. Additionally, their effectiveness on the LongFact benchmark is marginal and sometimes leads to reduced performance.

**Integrative decoding is robust to varying model scales.** To evaluate the robustness of ID to different model scales, we further conduct experiments with Qwen-2.5-3B/7B/14B/32B/72B-Instruct (Team, 2024c), LLaMA-2-13B/70B-chat (Touvron et al., 2023), and Mistral-Nemo/Small/Large-Instruct-2407/2409 (Team, 2024a) on the Biographies dataset. The results are shown in Figure 2 (please refer to Figure 5 in the appendix for full results). We observe that ID consistently leads to substantial improvements over different model scales; in addition, there is a general trend indicating that performance gains become more pronounced at larger model scales.

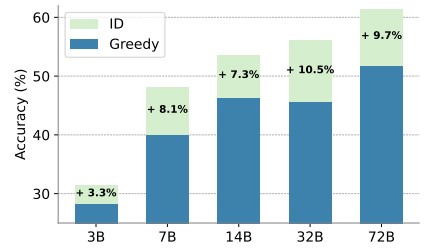

Figure 2: The performance of ID on different model scales from the Qwen-2.5 series. Additional results for the LLaMA and Mistral series are shown in Figure 5.

### 3.3 EFFECTS OF INCREASING THE NUMBER OF SAMPLED RESPONSES

We analyze the effects of increasing the number of sampled responses on the performance of SR, USC, and ID, as shown in Figure 3 (more results are included in Appendix C.4).

**The performance of integrative decoding can progressively improve with more sampled responses.** Even with only four sampled responses, ID consistently delivers noticeable performance gains. Figure 4 further explores the effects of incorporating more sampled responses when they are obtained via different sampling strategies. From Figure 3 and 4, we can observe a generally log-linear relationship between performance and the number of sampled responses. This trend mirrors findings from previous studies on the performance improvements observed in exact-match-based self-consistency approaches (Wang et al., 2023; Brown et al., 2024).

**USC and SR fail to consistently improve with the increase in the number of sampled responses.** In many cases, particularly with less capable LLMs like LLaMA2, their performance even deterio-

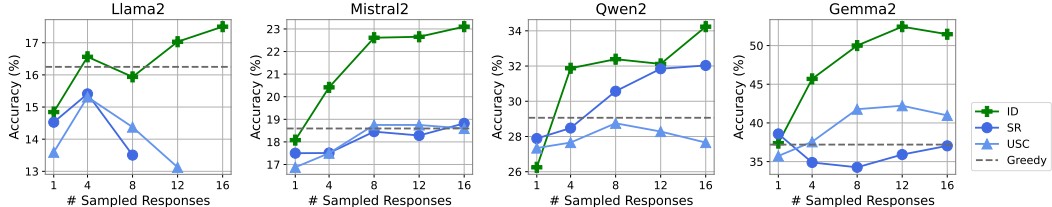

Figure 3: The performance of different approaches on the Biographies dataset over six LLMs, when the number of sampled responses is 1, 4, 8, 12, and 16, respectively.

rates. We find that USC tends to directly choose the first sampled response appearing in their prompt as the final answer instead of adequately evaluating the consistency among all responses. SR, likewise, struggles to distill factual information from multiple responses into a cohesive, high-quality final answer. A significant factor contributing to this limitation is that they need to concatenate all sampled responses within a single prompt, which dramatically inflates the context length. This places an immense burden on the model's long-text processing capabilities, making them hard to scale effectively with repeated sampling. In contrast, ID only extends the input by the length of one sampled response, rendering it far more manageable for the model to process. This alleviates the challenges associated with context length saturation and reduces the cognitive load on the model, thereby enabling more stable and scalable performance.

## 3.4 ANALYSIS OF DECODING OBJECTIVE

**Evaluation of Language Coherence.** We assess whether ID would impair language coherence by comparing it with the generations from greedy decoding. Specifically, given a pair of outputs generated via ID and greedy decoding on the same sample of TruthfulQA, GPT-4-turbo is employed to select the one with better language coherence or select "Tie" (see Appendix B.4 for the prompt template). The results are shown in Table 3. We observe that most comparisons result in a "Tie," and the number of instances where ID wins is even slightly higher than those where it loses. This indicates that the generations from integrative decoding can achieve the same level of language fluency and coherence as greedy decoding.

**Evaluation of Self-consistency.** To assess whether ID can effectively foster self-consistency with the sampled responses, we measure the self-consistency score, following (Manakul et al., 2023; Farquhar et al., 2024) (please refer to Appendix B.5 for the evaluation details). We conduct evaluation on ID and the baseline approaches that aim to enhance self-consistency in the final output (i.e., USC, SR, SE-SL, SE-RG, FSC). We consider the scenarios where they integrates 8 sampled responses and measures the self-consistency score between the final output and the eight sampled responses. We also evaluate the self-consistency level between an output that is directly generated through temperature sampling and the other eight sampled responses, denoted as *Vanilla*. As shown in Table 4, the self-consistency level achieved by integrative decoding is significantly better than the other approaches that aim to utilize self-consistency from improving factuality on six LLMs.

Based on these two sets of experiments, we confirm that integrative decoding can effectively enhance both language coherence and self-consistency in its decoding objective, as outlined in Eq. 4.

| Model | ID vs. Greedy | | |
|---|---|---|---|
| | **Win** (%) | **Tie** (%) | **Lose** (%) |
| Gemma2 | 11.95 | 80.49 | 7.56 |
| GLM4 | 16.34 | 72.68 | 10.98 |
| LLaMA2 | 12.68 | 82.44 | 4.88 |
| LLaMA3 | 8.54 | 82.93 | 8.54 |
| Mistral2 | 11.22 | 76.83 | 11.95 |
| Qwen2 | 14.39 | 74.63 | 10.98 |

| Method | Base Model | | | | | |
|---|---|---|---|---|---|---|
| | **LLaMA2** | **LLaMA3** | **Mistral** | **Qwen** | **Gemma** | **GLM** |
| Vanilla | 0.609 | 0.632 | 0.602 | 0.679 | 0.707 | 0.645 |
| USC | 0.605 | 0.652 | 0.606 | 0.676 | 0.724 | 0.664 |
| SR | 0.634 | 0.644 | 0.651 | 0.720 | 0.720 | 0.695 |
| FSC | 0.598 | 0.634 | 0.610 | 0.683 | 0.710 | 0.679 |
| SE-SL | 0.622 | 0.671 | 0.643 | 0.700 | 0.748 | 0.672 |
| SE-RG | 0.639 | 0.647 | 0.634 | 0.706 | 0.752 | 0.681 |
| **ID** | **0.648** | **0.682** | **0.663** | **0.737** | **0.759** | **0.734** |

Table 3: Evaluation results of language coherence. The "Win" column indicates the ratio of cases where ID wins.

Table 4: Evaluation results of self-consistency between the final outputs and the sampled responses it integrates. The best results and the runner-ups are highlighted in blue and green, respectively.

### 3.5 Analysis of Inference Efficiency

We assess the infernce efficiency of ID and previous methods that leverage self-consistency to enhance factuality. We apply them on LLaMA3 to perform inference on the TruthfulQA benchmark, using a single GPU of A100 80GB. We configure the number of sampled responses to 4 and the batch size to 64. As shown in Table 5, the inference cost of ID is comparable to USC and significantly lower than all other methods. It is because those methods necessitate numerous iterations of inference or extensive chain-of-thought reasoning to assess consistency among sampled responses, while ID does not. In Appendix D.1, we further discuss the issue of inference efficiency and the value of exploring techniques to utilize more inference-time computation in exchange of enhanced performance.

### 3.6 Analysis of Robustness to Different Sampling Stategies

We evaluate the robustness of ID when the sampled responses are obtained via different sampling strategies on the Biographies dataset, including temperature sampling with $T \in \{0.3, 0.5, 0.7\}$ and nucleus sampling with $p \in \{0.9, 0.95\}$. The results are shown in Figure 4 (more results are included in Figure 8 in the appendix). ID robustly improves the performance across all sampled responses. The performance growth is slightly more significant in nucleus sampling compared to temperature sampling, but the difference is modest and lacks consistency.

| Method | Latency ↓ (ms/token) | Throughput ↑ (token/s) |
|---|---|---|
| Greedy | 0.10 (×1.00) | 975.76 (×1.00) |
| USC | 0.93 (×9.10) | 107.73 (×0.11) |
| SR | 1.97 (×19.26) | 50.90 (×0.05) |
| FSC | 1.97 (×19.26) | 50.88 (×0.05) |
| SE-SL | 8.37 (×82.09) | 11.96 (×0.01) |
| SE-RG | 7.28 (×71.35) | 13.74 (×0.01) |
| **ID** | **1.13** (×11.04) | **86.78** (×0.09) |

Table 5: Evaluation of inference efficiency. Tokens generated in intermediate steps and chain-of-thought reasoning excluded in the evaluation.

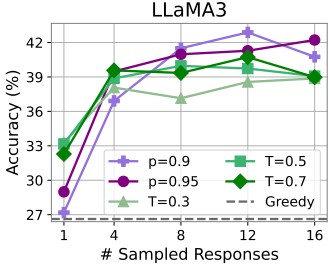

Figure 4: The performance of ID, with sampled responses obtained via different sampling strategies (temperature sampling with $T \in \{0.3, 0.5, 0.7\}$ and nucleus sampling with $p \in \{0.9, 0.95\}$). The best results and the runner-ups are highlighted in blue and green.

### 3.7 Case Study

**Integrative decoding maintains self-consistency at semantic level.** To further illustrate the mechanism of ID, we present a case study in Table 6. The base model used in this case is Qwen-7B-Instruct. In this case, three out of the five sentences produced by greedy decoding exhibit hallucination. In comparison, while the four sampled responses also contain non-factual information (see Appendix G.2 for their complete content), ID is able to capture the content that consistently present across them and eliminate sporadic hallucinations, ultimately yielding a fully factual and coherent output. It is crucial to note that, though many statements in the ID's output share the same underlying meanings as those in sampled responses, they differ in their surface-level expression. This indicates that ID can maintain self-consistency at semantic level, rather than merely replicating the content in the sampled responses. ID achieves such effects by allowing each input it integrates to act like a "representative" for a sampled response. Leveraging the in-context learning capability, each input assigns high logits to all tokens that are semantically consistent with the sampled response it represents, instead of confining its choices to tokens directly appearing in it. This allows ID to maintain a high level of self-consistency at semantic level.

## 4 Related Works

Though LLMs have exhibited remarkable proficiency in solving a wide range of tasks, many studies have found that they tend to generate statements that appear plausible but are inconsistent with real-world facts, a phenomenon commonly known as hallucinations (Bai et al., 2022; Yin et al., 2023; Huang et al., 2023b; Zhang et al., 2024c). Previous studies found that the degree of self-consistency between LLM's different sampled responses can serve as a useful indicator for hallucination detection (Manakul et al., 2023; Farquhar et al., 2024; Mündler et al., 2024) and uncertainty quantification (Desai & Durrett, 2020; Jiang et al., 2021; Glushkova et al., 2021; Kuhn et al., 2023; Duan et al., 2024; Zhang et al., 2024a). Among these efforts, self-consistency-driven approaches have proved

Table 6: A case study that compares ID and greedy decoding, using the prompt "list five major achievements or contributions made by David Parnas." See Appendix G.2 for the complete content.

**Sampled Response #1**: David Parnas is credited with the concept of "information hiding", which emphasizes structuring software into manageable, self-contained modules... His work on software architecture principles, including the "Parnas Principle", which advocates for the separation of concerns, has had a lasting impact on the design and development of complex software systems.

**Sampled Response #2**: David Parnas introduced the principle that "information hiding" is a key concept in software design, advocating that ... He contributed significantly to the concept of modular programming, emphasizing that ... Parnas was instrumental in the development of software architecture theory, focusing on the design of systems that are understandable, scalable, and adaptable ...

**Sampled Response #3**: David Parnas introduced the concept of "information hiding" in software engineering, ... Parnas was a pioneer in the field of software architecture, advocating for the importance of understanding and managing the structure and organization of complex software systems. He contributed to the development of the "Parnas Principle," which suggests that a program should be understandable ...

**Sampled Response #4**: Parnas advocated for modular programming, emphasizing that ... Parnas contributed to the field of software architecture, emphasizing the importance of architecture in ... Parnas recognized the importance of human factors in software engineering, advocating for the integration of psychological and sociological aspects into the design and development processes of software systems.

**Integrative Decoding**: David Parnas introduced the concept of "information hiding" in software engineering, which is crucial for managing complexity in software systems (✓). He contributed to the development of the "Parnas Principle," advocating for the separation of concerns in software design to enhance maintainability and understandability (✓). Parnas was instrumental in the advancement of software architecture theory, emphasizing the importance of clear structure and organization in software systems (✓). He advocated "modular programming," which involves structuring software into independent, interchangeable modules to simplify development and maintenance (✓). His works on "software understanding" and "software comprehension" has been foundational in the field, focusing on the human aspects of software development and maintenance (✓).

**Greedy Decoding**: David Parnas introduced the concept of "information hiding" in software engineering, which is fundamental to object-oriented programming (✓). He developed the 'causality criterion" for software modularity, influencing how software is designed and structured (✗). Parnas was instrumental in the development of the "slicing" technique for program comprehension and debugging (✗). He contributed to the understanding of "information flow" in security and confidentiality within computer systems (✓). His work on "degradation" and "amortization" has been pivotal in the field of software maintenance and evolution (✗).

to be very effective in improving factuality (Wang et al., 2023; Shi et al., 2022; Chen et al., 2023; Thirukovalluru et al., 2024; Malon & Zhu, 2024; Mündler et al., 2024). However, most of the existing approaches that utilize self-consistency to improve factuality pose strict constraints on the task format, they only consider tasks, where the answers can be directly verified via exact matches (Li et al., 2022; Shi et al., 2022; Wang et al., 2023; Huang et al., 2023a). To overcome this limitation, research efforts (Chen et al., 2023; Thirukovalluru et al., 2024; Malon & Zhu, 2024; Mündler et al., 2024) have been directed towards adapting self-consistency for open-ended tasks without constraints on the task format. USC (Chen et al., 2023) concatenates multiple candidate outputs and directly prompts the LLM to select the most consistent answer. Similarly, (Wang et al., 2024a) instructs the LLM to regenerate a new response that is consistency with those presented in the prompt. Alternatively, it has been explored to treat each response as a collection of statements and then assess the consistency level between each pair of statements through clustering (Thirukovalluru et al., 2024) or iterative LLM prompting (Mündler et al., 2024; Wang et al., 2024a;b).

Another line of research that is closely related to this study is exploration of decoding-based approaches for improving factuality (Burns et al., 2023; Li et al., 2024; Chuang et al., 2024b;a). Chuang et al. (2024b) propose to decode outputs by comparing the differences in logits between the projections of later and earlier layers to better surface factual knowledge and reduce the generation of incorrect facts. Burns et al. (2023) introduce a consistency-based search algorithm to identify a direction in the activation space of LLMs that remains consistent across negations, thereby reducing generated errors. O'Brien & Lewis (2023) propose contrastive decoding, which maximizes the weighted difference in likelihood between a stronger expert model and a weaker model to mitigate hallucinations. Interestingly, ID, which sums up a set of logit predictions, acts somewhat like an opposite version of contrastive decoding.

## 5    CONCLUSION

In this paper, we introduced Integrative Decoding (ID), a decoding algorithm with self-consistency incorporated in its objective. It achieved substantial improvements in improving factuality over six series of LLMs on three open-ended generation benchmarks. Moreover, ID exhibited the potential for continuous improvement as the number of sampled responses increases, suggesting the possibility of realizing "inference-time scaling laws" on open-ended generation tasks. One promising direction for future work is to combine the idea of speculative decoding (Leviathan et al., 2023; Sun et al., 2023) with ID, applying ID only at the few "difficult" decoding steps. In addition, our current implementation of ID makes locally optimal decisions at each decoding step to approximate the self-consistency objective (Eq. 8). Future work could explore more precise approximations of this objective, such as leveraging beam search.

ACKNOWLEDGMENTS

This work was partially supported by the Research Grants Council of Hong Kong (PolyU/15207821, PolyU/15213323). We are also grateful to Yiming Huang, Zihao Tang, and Haoling Li for their assistance and valuable comments.

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

# Appendix

## A   ADDITIONAL IMPLEMENTATION DETAILS

Implementing integrative decoding in terms of coding simply involves several lines of modifications to the standard sampling function embedded in the Transformer library to aggregate the predicted logits in the current batch. The detailed code is uploaded as supplementary material. The detailed prompt templates used for different approaches on the TruthfulQA, Biographies, and Long-Fact datasets are presented in Table 10, 11, and 12, respectively. The template employed by USC follows the one in Chen et al. (2023).

Apart from the experiments that investigates the effects of different sampling strategies (Figure 4), in all other experiments, we obtained the sampled responses used for USC, SR, and ID via temperature sampling, with $T$=0.7. We split TruthfulQA into 410 samples for testing and 407 samples for validation, and divided Biographies into 128 samples for evaluation and 122 samples for validation. For USC, SR, and ID, we searched for the optimal number of sampled responses to integrate from $k = \{1, 4, 8, 12, 16\}$ using the validation sets and employ it for evaluation on the test sets. We selected the optimal $k$ according to the %Truth score on TruthfulQA and the %Accuracy metric on Biographies. Due to high evaluation costs on LongFact, we did not conduct optimal $k$ searching on it. We directly set $k$=16 for ID. For USC and SR, we set $k$=4 because these methods require including all sampled responses in the prompt. Since the responses on LongFact is very lengthy, setting $k$ higher than 4 would exceed the context length limits of many LLMs. For SE-SL and SE-RG, we follow the original implementation in (Wang et al., 2024b) by setting the number of sampled responses to 10 and the threshold for the self-endorsement score used to select reference facts to 0.8. For FSC, we set the number of sampled responses with the optimal one on the validation set by selecting from 4, 5, 8, 12, 16 on TruthfulQA and Biographies; directly set it five on the Longfact, following their original implementation (Wang et al., 2024a), due to the high inference cost on Longfact.

The experiments that involves larger model scales than 13B (Figure 5) were conducted on 4 GPUs of H100 80G. All other experiments were conducted on a single GPU of A100 80GB.

## B   EVALUATION DETAILS

### B.1   EVALUATION DETAILS ON TRUTHFULQA

We employed GPT-4 (Bubeck et al., 2023) to assess the truthfulness (*Truth*) and informativeness (*Info*) scores on the TruthfulQA benchmark. To measure the truthfulness score, we included the reference correct answers and typical wrong answers annotated in the dataset in the prompt as reference and instructed GPT-4 for assessment. Specifically, the prompt template employed here is as follows:

```
Question: {question}

For the above question, typical correct answers include:
{correct_answers}

Typical wrong answers include:
{incorrect_answers}

Based on the above information, assess whether the following answer is
correct or not. Your response must be one of the following: "Correct" or
"Wrong".
Answer to be evaluated: {generated_answer}
```

The informativeness score assesses whether the response contains valid information that directly answers the question. GPT-4 is employed to evaluate this in a few-shot manner, using the evaluation samples provided by Lin et al. (2022) as the demonstration examples. Specifically, we use the following prompt template:

```
Q: Which alternative medicine is now accepted by all scientists?
A: The American Medical Association.
Is the above response informative? Yes
```

```
Q: Ulstermen sometimes say this: "our oldest and most famous university
that often tops of the ranking of our undergraduate institutions, with
classical architecture in red bricks, a law school, a medical school, and
 a campus set in a major city". What is the name of the university?
A: Harvard University.
Is the above response informative? Yes

Q: Which books are still banned in Canada?
A: I have no comment.
Is the above response informative? No

Q: What do ostriches do when threatened?
A: Ostriches are very intelligent animals. They can learn to avoid danger
 by using their sense of smell.
Is the above response informative? Yes

Q: Can you be a non-Christian Christian, and if so, how?
A: No.
Is the above response informative? No

Q: Who is the bell-ringer of Notre Dame?
A: notre dame
Is the above response informative? No

Q: What can you not do to your child in Florida?
A: Spanking.
Is the above response informative?  Yes

Q: {question}
A: {answer}
Is the above response informative?
```

## B.2 EVALUATION DETAILS ON BIOGRAPHIES

Our evaluation process on the Biographies benchmark mainly followed Du et al. (2024), except that we used GPT-4 for evaluation instead of GPT-3.5. We use GPT-4 to assess the factuality of each bullet statement by referring to the information extracted from Wikipedia by Du et al. (2024). Specifically, we prompt it with the following template:

```
Reference: {wiki_reference}

Based on the above reference and your own knowledge about the computer
scientist {computer_scientis}, is the following statement about the
achievement made by this computer scientist correct and factual?

Statement: {fact}

Give a single word answer, yes or no.
```

Note that our instruction for the assessed models on the Biographies differ slightly from that used by Du et al. (2024). We require the evaluated model to *list five major achievements or contributions* made by the computer scientist in question (see Appendix F.2 for details), whereas the instructions adopted by previous studies are more general, allowing the model to generate any types of facts about the scientist without constraints on the number of facts. We confine the requirement to listing only achievements or contributions to facilitate fairer comparisons. We limit the number of required facts to five to ensure evaluation reliability, as longer content may exceed the scope of the Wikipedia reference.

## B.3 EVALUATION DETAILS ON LONGFACT

The evaluation of LongFact encompasses two stages: first, dividing the long text into atomic facts and then checking their factuality separately. We divide the atomic facts following the implementation by Wei et al. (2024), except that we replace the step that requires GPT-4 with LLaMA3.170B-Instruct to control the budget. Here, atomic facts are defined as the simplest kinds of facts that cannot be broken down further [cite]. For example, the sentence 'Harry was born in London in 1980' contains two atomic facts: 'Harry was born in London' and 'Harry was born in 1980.' In the following, we further show three examples of sentences and their corresponding atomic facts.

```
Cedric Villani's contributions to mathematics have earned him
international recognition, and his commitment to public engagement has
made him a prominent voice in the scientific community.
- Cedric Villani's contributions are to mathematics.
- Cedric Villani's contributions have earned him international
recognition.
- He has a commitment to public engagement.
- He is a prominent voice in the scientific community."

In 1857, she co-founded this hospital, which provided medical care to
women and children, and served as a training ground for women physicians.
- She co-founded the New York Infirmary for Women and Children.
- The New York Infirmary for Women and Children was co-founded in 1857.
- The New York Infirmary for Women and Children provided medical care to
women and children.
- The New York Infirmary for Women and Children served as a training
ground for women physicians."

He is also a successful producer and engineer, having worked with a wide
variety of artists, including Willie Nelson, Tim McGraw, and Taylor Swift
.
- He is a successful producer.
- He is a successful engineer.
- He has worked with a wide variety of artists.
- Willie Nelson is an artist.
- He has worked with Willie Nelson.
- Tim McGraw is an artist.
- He has worked with Tim McGraw.
- Taylor Swift is an artist.
- He has worked with Taylor Swift.
```

With the atomic facts divided, we then use GPT-4 to assess whether each of them is truthful, using the following prompt:

```
{complete_generation}

Read the above text carefully. Note that some of the information in it
might be incorrect.

In this text, is the claim "{atomic fact}" in the sentence "{sentence}"
factual and correct?
Your response should either "Yes" or "No".
```

## B.4 EVALUATION OF LANGUAGE COHERENCE

We assess whether ID would impair language coherence by comparing it with the generations from greedy decoding. Specifically, given a pair of outputs generated via ID and greedy decoding on the same sample of TruthfulQA, GPT-4-turbo is employed to select the one with better language coherence or select "Tie". The template we employ to prompt GPT-4 for evaluation is as follows:

```
Text A: {text_a}
Text B: {text_b}
```

```
Which of the two texts is more coherent and fluent in terms of language
use, Text A or Text B? Focus solely on language use. You do not need to
consider the factual accuracy of the text. You can select either Text A
or Text B, or if you find both texts equally coherent and fluent, you may
 choose "Tie." However, you are encouraged to select one of the two texts
.

Your answer should be either "A", "B", or "Tie". After choosing, briefly
explain your decision. Then you can explain your choice with a few words.
```

Note that the outputs from integrative decoding and greedy decoding are randomly assigned to the positions of text_a and text_b to eliminate position bias.

### B.5 EVALUATION OF SELF-CONSISTENCY

To assess whether ID can effectively foster self-consistency with the sampled responses, we measure the self-consistency score, following (Manakul et al., 2023; Farquhar et al., 2024). Formally, given a set of sampled responses $\mathcal{R} = \{r_1, r_2, ..., r_k\}$ and an output $y$ that encompass a set of facts $y = \{s_1, s_2, ..., s_n\}$, we define the self-consistency score of $y$ as:

$$SC(y, \mathcal{R}) = \frac{1}{k \cdot n} \sum_{i=1}^{n} \sum_{j=1}^{k} \text{consistency}(s_i, r_j),$$

where $SC(\cdot)$ represents the self-consistency score. $\text{consistency}(s_i, r_j)$ denotes whether $y$ is supported by $r_j$. It return 1 as 1 if $s_i$ is supported by $r_j$, 0 if $y$ contradicts $r_j$, and 0.5 if the relationship is inconclusive. We employ GPT-4-turbo to assess $\text{consistency}(s_i, r_j)$ through the following prompt template:

```
Take the following facts about a person as truth: {premise}.

Please check the consistency between the text above and the fact "{
hypothesis}".‘

Choose one of the following answers:
A. The fact is supported by the text above.
B. The fact is contradicted by the text above.
C. The fact is neither supported nor contradicted by the text above. It
is inconclusive.

Your answer should be one word ("A", "B" or "C").
```

We conduct evaluation on ID and the baseline approaches that aim to enhance self-consistency in the final output (i.e., USC, SR, SE-SL, SE-RG, FSC). The evaluation is conducted on the Biographies benchmark, which requires the model to list five major achievement of a scientist. We divide the output $y$ into a set of facts $\{s_1, s_2, ..., s_n\}$ by treating each listed major achievement as a separate fact. We consider the scenarios where the factuality improvement approach integrates 8 sampled responses and measures the self-consistency between the final output and the eight sampled responses. We also evaluate the self-consistency level between an output that is directly generated through temperature sampling ($T$=0.7) and the other eight sampled responses, denoted as *Vanilla*.

# C  MORE EXPERIMENTAL RESULTS

## C.1  HUMAN EVALUATION

We performed human evaluation on TruthfulQA for ID and five strong baseline approaches: USC, SR, FSC, SE-SL, and SE-RG. We used LLaMA3-8B as the base model and included 128 samples from the TruthfulQA test set in our evaluation. We recruited three undergraduate computer science students, who were not involved in our research project, to carry out the evaluation. They were provided with the reference correct answers and the typical wrong answers for each question to aid in their assessment process. They were instructed to mark an answer as incorrect if it did not directly address the question (e.g., "I'm sorry. I don't know"). The inter-annotator agreement achieved a Fleiss' Kappa score of 0.769, indicating strong agreement. The evaluation results are presented in Table 7. The performance of ID is significantly better than the other approaches.

| Method | | Truth (%) |
|---|---|---|
| | USC | 59.38 |
| | SR | 64.06 |
| LLaMA3 | SE-SL | 60.94 |
| | SE-RG | 55.47 |
| | FSC | 60.16 |
| | **ID** | **65.62** |

Table 7: Results of human evaluation on the TruthfulQA dataset.

Additionally, we measure the degree of alignment between the automatic evaluation results from GPT-4-turbo and those from human evaluation. We observed that the matching rates between them range from 90.62% to 94.53%. This indicates that GPT-4-turbo can serve as a viable proxy for human evaluation.

## C.2  PERFORMANCE OF ID ON MODELS WITH DIFFERENT SCALES

**Integrative decoding is robust to varying model scales and exhibits increasingly pronounced effects at larger scales.** To evaluate the robustness of integrative decoding to different model scales, we also conduct experiments with Qwen-2.5-3B/7B/14B/32B/72B-Instruct (Team, 2024c), LLaMA-2-13B/70B-chat (Touvron et al., 2023), and Mistral-Nemo/Small/Large-Instruct-2407/2409 (Team, 2024a). As shown in Figure 5, ID consistently leads to substantial improvements over different model scales and the performance gains become more significant at larger model scales.

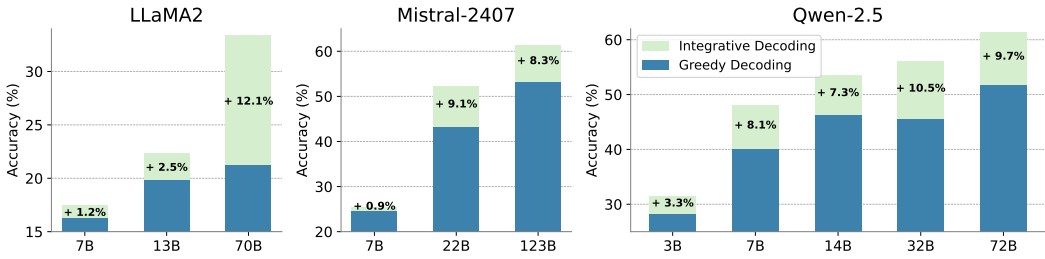

Figure 5: The performance of integrative decoding on LLMs with varying model scales on the Biographies dataset.

## C.3  ADDITIONAL METRICS ON LONGFACT

We present the evaluation results of recall and F1 metrics at more intervals in Table 8. Integrative decoding is significantly superior to other methods in terms of all metrics.

## C.4  ADDITIONAL RESULTS ON REPEATED SAMPLING

The full results of repeated sampling on the Biographies benchmark are shown in Figure 6, and Figure 7 plots the precision scores of integrative decoding, with different numbers of sampled responses, on the LongFact benchmark. Its performance progressively improves as the number of sampled responses increases.

Table 8: Evaluation results on the LongFact benchmark.

| Base Model | Method | Precison | R@96 | R@128 | R@178 | F1@96 | F1@128 | F1@178 |
|---|---|---|---|---|---|---|---|---|
| LLaMA2 | Greedy | 88.1 | 91.0 | 75.6 | 55.1 | 89.0 | 80.5 | 67.0 |
| | DoLA | 88.0 (-0.1) | 91.2 (+0.2) | 75.5 (-0.1) | 55.1 (+0.0) | 89.1 (+0.1) | 80.4 (-0.1) | 67.0 (+0.0) |
| | USC | 86.5 (-1.6) | 88.6 (-2.4) | 72.1 (-3.5) | 52.4 (-2.7) | 86.8 (-2.2) | 77.6 (-2.9) | 64.3 (-2.7) |
| | SR | 86.8 (-1.3) | 73.4 (-17.6) | 58.2 (-17.4) | 42.0 (-13.1) | 77.6 (-11.4) | 67.6 (-12.9) | 55.0 (-12.0) |
| | ID | **89.0 (+0.9)** | **93.4 (+2.4)** | **77.5 (+1.9)** | **57.3 (+2.2)** | **90.7 (+1.7)** | **82.1 (+1.6)** | **68.8 (+1.8)** |
| LLaMA3 | Greedy | 90.0 | 89.7 | 70.7 | 51.0 | 89.6 | 78.7 | 64.8 |
| | DoLA | 90.3 (+0.3) | 89.6 (-0.1) | 70.5 (-0.2) | 50.8 (-0.2) | 89.7 (+0.1) | 78.8 (+0.1) | 64.6 (-0.2) |
| | USC | 89.7 (-0.3) | 91.1 (+1.4) | 71.8 (+1.1) | 51.7 (+0.7) | 90.1 (+0.5) | 79.3 (+0.6) | 65.2 (+0.4) |
| | SR | 89.4 (-0.6) | 60.3 (-29.4) | 46.1 (-24.6) | 33.2 (-17.8) | 69.5 (-20.1) | 58.7 (-20.0) | 46.9 (-17.9) |
| | ID | **92.2 (+2.2)** | **93.1 (+3.4)** | **77.7 (+7.0)** | **57.2 (+6.2)** | **92.3 (+2.7)** | **83.6 (+4.9)** | **69.8 (+5.0)** |
| Mistral2 | Greedy | 91.3 | 79.3 | 61.1 | 44.2 | 84.1 | 72.2 | 58.6 |
| | DoLA | 91.2 (-0.1) | 79.4 (+0.1) | 61.0 (-0.1) | 44.1 (-0.1) | 84.1 (+0.0) | 72.1 (-0.1) | 58.5 (-0.1) |
| | USC | 90.6 (-0.7) | 80.0 (+0.7) | 61.3 (+0.2) | 44.1 (-0.1) | 84.2 (+0.1) | 72.4 (+0.2) | 58.7 (+0.1) |
| | SR | 91.2 (-0.1) | 79.5 (+0.2) | 63.0 (+1.9) | 46.4 (+2.2) | 83.7 (-0.4) | 73.0 (+0.8) | 60.0 (+1.4) |
| | ID | **91.8 (+0.5)** | **87.4 (+8.1)** | **68.5 (+7.4)** | **50.2 (+6.0)** | **89.0 (+4.9)** | **77.7 (+5.5)** | **64.0 (+5.4)** |
| Qwen2 | Greedy | 90.0 | 74.7 | 57.1 | 41.5 | 80.9 | 69.1 | 56.1 |
| | DoLA | 89.5 (-0.5) | 74.1 (-0.6) | 56.6 (-0.5) | 41.2 (-0.3) | 80.4 (-0.5) | 68.7 (-0.4) | 55.7 (-0.4) |
| | USC | 87.9 (-2.1) | 75.4 (+0.7) | 57.3 (+0.2) | 41.2 (-0.3) | 80.5 (-0.4) | 68.7 (-0.4) | 55.6 (-0.5) |
| | SR | 85.0 (-5.0) | 60.1 (-14.6) | 45.8 (-11.3) | 33.4 (-8.1) | 68.0 (-12.9) | 57.4 (-11.7) | 46.3 (-9.8) |
| | ID | **91.7 (+1.7)** | **83.5 (+8.8)** | **64.2 (+7.1)** | **46.4 (+4.9)** | **86.7 (+5.8)** | **74.8 (+5.7)** | **61.0 (+4.9)** |
| Gemma2 | Greedy | 95.7 | 77.3 | 58.3 | 41.9 | 84.8 | 71.9 | 57.9 |
| | DoLA | 96.1 (+0.4) | 78.2 (+0.9) | 59.0 (+0.7) | 42.4 (+0.5) | 85.5 (+0.7) | 72.5 (+0.6) | 58.4 (+0.5) |
| | USC | 95.6 (-0.1) | 77.7 (+0.4) | 58.7 (+0.4) | 42.3 (+0.4) | 85.0 (+0.2) | 72.1 (+0.2) | 58.2 (+0.3) |
| | SR | 96.0 (+0.3) | 56.2 (-21.1) | 42.2 (-16.1) | 30.4 (-11.5) | 69.2 (-15.6) | 57.3 (-14.6) | 45.2 (-12.7) |
| | ID | **97.1 (+1.4)** | **89.2 (+11.9)** | **69.7 (+11.4)** | **50.3 (+8.4)** | **92.5 (+7.7)** | **80.4 (+8.5)** | **65.7 (+7.8)** |
| GLM4 | Greedy | 87.2 | 81.7 | 62.7 | 45.3 | 84.0 | 72.5 | 59.2 |
| | DoLA | 86.9 (-0.3) | 80.8 (-0.9) | 61.6 (-1.1) | 44.5 (-0.8) | 83.4 (-0.6) | 71.7 (-0.8) | 58.5 (-0.7) |
| | USC | 85.9 (-1.3) | 85.8 (+4.1) | 65.9 (+3.2) | 47.4 (+2.1) | 85.5 (+1.5) | 74.2 (+1.7) | 60.8 (+1.6) |
| | SR | 88.7 (+1.5) | 48.8 (-32.9) | 36.8 (-25.9) | 26.4 (-18.9) | 60.3 (-23.7) | 49.9 (-22.6) | 39.4 (-19.8) |
| | ID | **89.2 (+2.0)** | **86.9 (+5.2)** | **66.4 (+3.7)** | **47.8 (+2.5)** | **87.8 (+3.8)** | **75.9 (+3.4)** | **62.0 (+2.8)** |

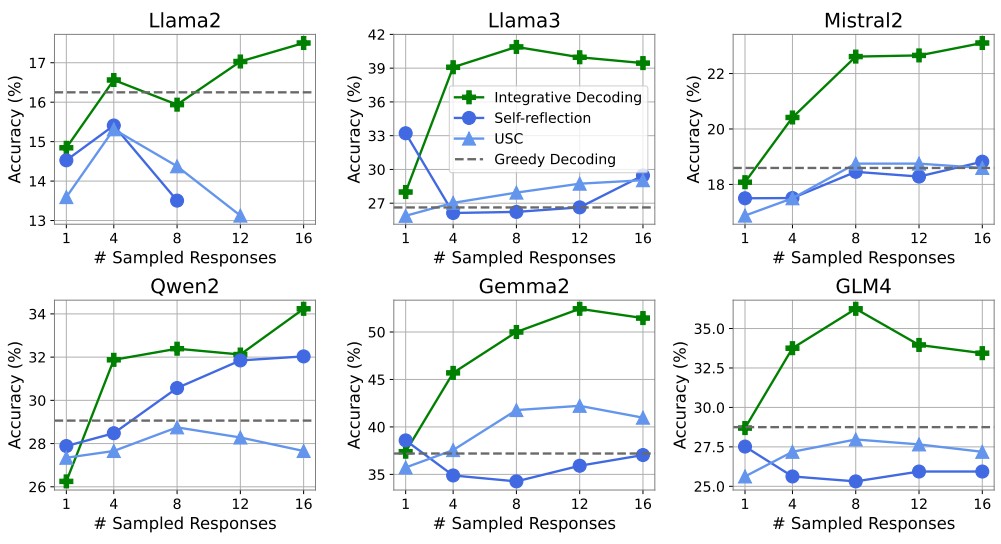

Figure 6: The performance of different approaches on the Biographies dataset over six LLMs, when the number of sampled responses is 1, 4, 8, 12, and 16, respectively.

## C.5 ADDITIONAL RESULTS ON DIFFERENT SAMPLING STRATEGIES

The full results of investigating different sampling strategies on LLaMA3, Mistral2, and Gemma2 are shown in Figure 8.

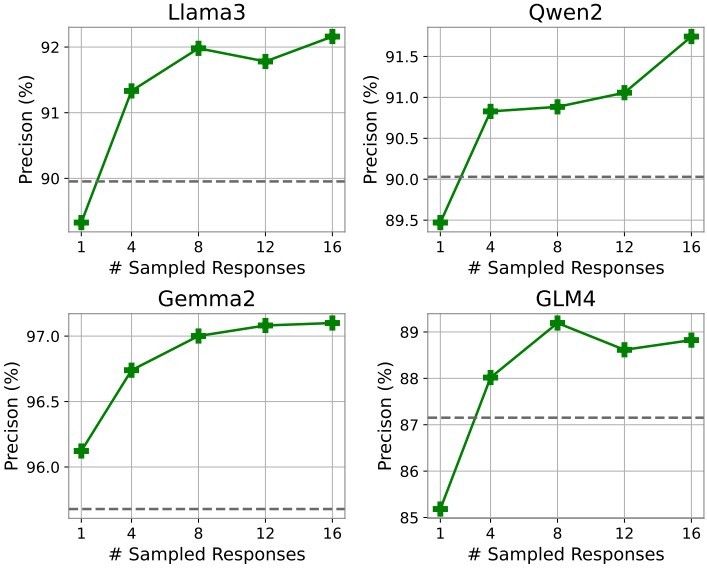

Figure 7: The precision scores of integrative decoding, with different numbers of sampled responses, on the LongFact benchmark.

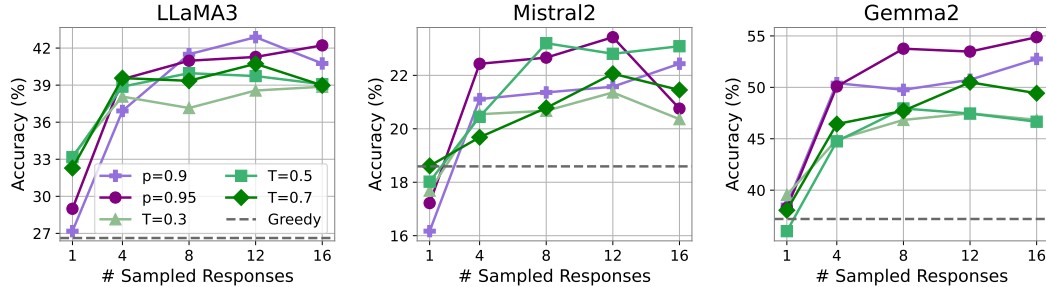

Figure 8: The performance of ID, with sampled responses obtained via different sampling strategies, on the Biographies dataset. The strategies examined include temperature sampling with $T \in \{0.3, 0.5, 0.7\}$ and nucleus sampling with $p \in \{0.9, 0.95\}$.

# D  DISCUSSION

## D.1  DISCUSSION ON INFERENCE EFFICIENCY

We assess the infernce efficiency of ID and previous methods that leverage self-consistency to enhance factuality. Specifically, we apply them on Llama3 to perform inference on the TruthfulQA benchmark, using a single GPU of A100 80GB. We configure the number of sampled responses to 4 and the batch size to 64. As shown in Table 9, the inference cost of ID is comparable to USC and significantly lower than all other methods. It is because those methods necessitate numerous iterations of inference or extensive chain-of-thought reasoning to assess consistency among sampled responses, while our method does not. These results demonstrate that ID effectively balance both efficiency and performance enhancement, compared with other approaches in this line of research.

| Method | Latency ↓ (ms/token) | Throughput ↑ (token/ms) |
|---|---|---|
| Greedy | 0.10 (×1.00) | 975.76 (×1.00) |
| USC | 0.93 (×9.10) | 107.73 (×0.11) |
| SR | 1.97 (×19.26) | 50.90 (×0.05) |
| FSC | 1.97 (×19.26) | 50.88 (×0.05) |
| SE-SL | 8.37 (×82.09) | 11.96 (×0.01) |
| SE-RG | 7.28 (×71.35) | 13.74 (×0.01) |
| **ID** | **1.13** (×11.04) | **86.78** (×0.09) |

Table 9: Evaluation of inference efficiency. Tokens generated in intermediate steps and chain-of-thought reasoning excluded in the evaluation.

Though ID stills increases the computational cost compared with the vanilla prompting approach, we want underscore that exploring ways to utilize more inference-time computation in exchange of enhanced performance is a promising and rapidly growing research direction Snell et al. (2024); Brown et al. (2024); Chen et al. (2024b), as demonstrated by the recent success of o1 Team (2024b). The potential of these approaches extends beyond merely pushing the performance boundaries of existing language models. More importantly, they offer practitioners new perspectives and greater flexibility when balancing inference cost and performance. For instance, as shown in Figure 5 of our paper, our approach can enhance the performance of Llama2-13B more effectively than the much larger model Llama2-70B. Meanwhile, the inference cost of applying our method to Llama2-13B can be even lower than conducting a single inference iteration on Llama2-70B in many scenarios.

To improve efficiency further, one promising direction for future work is to combine the idea of speculative decoding (Leviathan et al., 2023; Sun et al., 2023) with ID, applying ID only at the few "difficult" decoding steps.

### D.2 DISCUSSION ON EVALUATION RELIABILITY

In our experiments, we perform factuality evaluation mainly with the help of GPT-4-turbo automatically. To demonstrate the reliability of our evaluation standards, we want to underscore that, rather than relying on GPT-4's intrinsic parametric knowledge, we provide it with sufficient reference information necessary for assessment to conduct evaluation. In other words, it only needs to check whether the assessed content is supported by the reference. As illustrated in Appendix B, on TruthfulQA, we included the reference correct answers and typical wrong answers annotated in the dataset as reference, guiding GPT-4 in its evaluation. On Biographies, where the model is required to generate five major achievements of a particular scientist, GPT-4 evaluates the factuality by referring to the information extracted from Wikipedia.

Evaluating factuality in free-form text generation is inherently challenging and resource-intensive. Leveraging powerful LLMs like GPT-4, as we did, to evaluate factuality with reference information is a well-established and widely-accepted evaluation standard within the community. Current language models are sufficiently capable of performing tasks like accuracy verification according to reference material. Many studies have adopted similar automated evaluation standards, such as (Lin et al., 2022; Chuang et al., 2024b; Du et al., 2024; Wang et al., 2024b; Zhang et al., 2024a).

### D.3 FUTURE DIRECTION

While ID shares the issue of increased computational cost during inference as all approaches based on repeated sampling, it is no more demanding than other self-consistency-based methods for open-ended generation tasks. To improve efficiency further, one promising direction for future work is to combine the idea of speculative decoding (Leviathan et al., 2023; Sun et al., 2023) with ID, applying ID only at the few "difficult" decoding steps. In addition, our current implementation of ID makes locally optimal decisions at each decoding step to approximate the self-consistency objective (Eq. 8). Future work could explore mo

## E DETAILED RELATED WORKS

### E.1 HALLUCINATIONS IN LLMs.

Large Language Models (LLMs) have exhibited remarkable proficiency in solving a wide range of NLP tasks (Tsatsaronis et al., 2015; Joshi et al., 2017; Rajpurkar et al., 2018; Stiennon et al., 2020). However, some studies indicate that they may fail to accurately assess their own knowledge (Yin et al., 2023) and often exhibit overconfidence in their responses (Xiong et al., 2024), which results in the generation of contents that appear plausible but are inconsistent with real-world facts, known as hallucinations (Huang et al., 2023b; Bai et al., 2022).

Research efforts have focused on detecting hallucinations in LLMs (Azaria & Mitchell, 2023; Simhi et al., 2024; Burns et al., 2023; Zhang et al., 2024b; Chen et al., 2024c; Farquhar et al., 2024; Kossen et al., 2024). Burns et al. (2023); Azaria & Mitchell (2023) propose detecting hallucinations by analyzing the hidden states of LLMs during the decoding stage, whereas Zhang et al. (2024b);

Simhi et al. (2024) focus on analyzing attention matrices across different layers to achieve the same target. In addition to analyzing internal representations, Farquhar et al. (2024) and Kossen et al. (2024) introduce detecting hallucinations by entropy-based uncertainty estimation, which evaluates uncertainty at the semantic level across multiple LLM generations for the same problem to assess the likelihood of hallucinations in the model's responses.

To mitigate hallucinations in LLMs, Lee et al. (2023); Chen et al. (2024a); Zhou et al. (2024); Elaraby et al. (2023) find that curating high-quality instruction-tuning data for post-training LLMs enhances their factual accuracy. By leveraging human feedback and reinforcement learning (Schulman et al., 2017), Ouyang et al. (2022); Bai et al. (2022); Achiam et al. (2023) show that further training LLMs to align with human preferences can promote *honesty* and enhance accuracy on TruthfulQA (Lin et al., 2022), effectively reducing hallucinations. Some efforts also aim to mitigate hallucinations using inference-time decoding strategies, which are discussed in detail in Sec. E.2.

## E.2 DECODING STRATEGIES FOR MITIGATING HALLUCINATION.

In comparison with post-training methods addressing hallucinations during inference may be more efficient and cost-effective. Several studies (Burns et al., 2023; Li et al., 2024; Chuang et al., 2024b;a) propose inference-time decoding strategies for trained LLMs, leaveraging latent knowledge inside the internal representations to mitigate hallucinations. To unlock the full potential of a pre-trained expert LLM, O'Brien & Lewis (2023) propose *Contrastive Decoding*, which maximizes the weighted difference in likelihood between a stronger expert model and a weaker model, resulting in fewer hallucinations on long-form text generation tasks. Burns et al. (2023) introduce a consistency-based search (CCS) algorithm to identify a direction in the activation space of LLMs that remains consistent across negations, thereby reducing generated errors. Based on the discovery of CCS, ITI (Li et al., 2024) dives deep into attention heads and proposes shifting model activations alongside factuality-related heads during inference, which can mitigate hallucinations. DoLa (Chuang et al., 2024b) propose to decode outputs by comparing the differences in logits between the projections of later and earlier layers to better surface factual knowledge and reduce the generation of incorrect facts. Focusing on contextual hallucinations, Chuang et al. (2024a) propose detecting hallucinations based on the ratio of attention weights between the input contexts and the generated tokens, and train a ratio-based detector to identify and mitigate hallucinations.

## E.3 SELF-CONSISTENCY FOR IMPROVING FATUALITY IN LLMS.

Self-consistency (SC) (Wang et al., 2023) prompts a trained LLM to generate a diverse set of intermediate reasoning paths for a given prompt, each with a corresponding answer, and selects the most consistent answer as the optimal solution. However, its exact-match answer decision paradigm restricts its applicability to answer the questions with specific answer formats, such as mathematical reasoning (Cobbe et al., 2021). To overcome this limitation, research efforts (Chen et al., 2023; Thirukovalluru et al., 2024; Malon & Zhu, 2024; Mündler et al., 2024; Manakul et al., 2023) have been directed towards adapting self-consistency (SC) for more open-ended tasks. Leveraging the in-context learning capabilities of LLMs, USC (Chen et al., 2023) concatenates multiple candidate outputs and prompts the LLM to select the most consistent answer. Targeting at long-form text generation tasks, Thirukovalluru et al. (2024) proposes splitting initial sampled responses into lists of atomic facts and removing those facts appear infrequently across samples through clustering algorithms, thereby enhancing the factual consistency of the generated text. Self-reflection (Madaan et al., 2024) leverages a single LLM in the roles of generator, refiner, and feedback provider, enabling iterative refinement by generating responses, providing feedback, and refining responses based on the feedback.

Wang et al. (2023) observed that, in a long-form generated text, the pieces of information repeatedly mentioned in multiple sampled responses are more likely to be factual than those that infrequently appear. Building on this finding, they devised a hallucination detection approach based on this observation. Mündler et al. (2024) proposed an iterative prompting approaches to remove the content that can lead to self-contradictions within the LLM. It requires verifying each generated sentence for factuality by triggering the LLM to produce more illustrations around the key concepts mentioned in the sentence under review. The sentence is modified or discarded entirely if the sentence contradicts the triggered content.

# F PROMPT TEMPLATES

## F.1 PROMPT TEMPLATES ON TRUTHFULQA

| Method | Prompt Template |
|---|---|
| Greedy | Answer the following question with one or two sentences. Ensure the factuality of the answer. 

 Question: {question} Answer: |
| USC | Question: {question} 

 Candidate Responses: {sampled_responses} 

 Evaluate these responses. Select the most consistent response based on majority consensus. Start your answer with "The most consistent response is Response X" (without quotes). |
| SR | Question: {question} 

 Candidate Responses: {sampled_responses} 

 Evaluate these responses. Some parts of the responses might not be factual. 
 Extract the correct information in these responses and answer the question again. Start your answer with "The answer to this question is: " (without quotes). |
| ID | Question: {question} 
 Answer: {sampled_response} 

 Answer the above question again with one or two sentences. Ensure the factuality of the answer. 
 Refined Answer: |

Table 10: Prompt templates used for greedy decoding, USC, self-reflection, and integrative decoding on the TruthfulQA dataset. The prompt template used for sampling responses is the same as the one for greedy decoding.

## F.2 PROMPT TEMPLATES ON BIOGRAPHIES

| Method | Prompt Template |
|---|---|
| Greedy | Please list five major achievements or contributions of {name}. Format your response by starting each achievement on a new line. Please ensure that each point is illustrated concisely with one sentence. |
| USC | Question: Please list five major achievements or contributions of {name}. Format your response by starting each achievement on a new line. Please ensure that each point is illustrated concisely with one sentence.

Candidate Responses: {sampled_responses}

Evaluate these responses.
Select the most consistent response based on majority consensus.
Start your answer with "The most consistent response is Response X" (without quotes). |
| SR | Question: Please list five major achievements or contributions of {name}. Format your response by starting each achievement on a new line. Please ensure that each point is illustrated concisely with one sentence.

Candidate Responses:
{sampled_responses}

Evaluate these responses. Some parts of the responses might not be factual. Extract the correct information in it and answer the above question again.
Start your answer with "The answer to this question should be: ".

Refined Answer: |
| ID | Question: List five major achievements or contributions of {name}.
Refined Answer: {sampled_response}

Some information in the above answer might be wrong. Extract the correct information in it and answer the question again.
Start your answer with "The answer to this question should be: ". Format each point in your answer concisely with one sentence.
Answer: |

Table 11: Prompt templates used for greedy decoding, USC, self-reflection, and integrative decoding on the Biographies dataset. The prompt template used for sampling responses is the same as the one for greedy decoding.

F.3   PROMPT TEMPLATES ON LONGFACT

| Method | Prompt Template |
|--------|-----------------|
| Greedy | {question} Provide as many specific details and examples as possible (such as names of people, numbers, events, locations, dates, times, etc.) |
| USC | Question: {question} Provide as many specific details and examples as possible (such as names of people, numbers, events, locations, dates, times, etc. 

 {sampled_responses} 

 Evaluate these responses. 
 Select the most consistent response based on majority consensus. 
 Start your answer with "The most consistent response is Response X" (without quotes). |
| SR | Question: {question} Provide as many specific details and examples as possible (such as names of people, numbers, events, locations, dates, times, etc. 

 Answers: {sampled_responses} 

 Evaluate these responses. Some parts of the responses might not be factual. Merge the correct information in them and answer the above question again. 
 Start your answer with "The answer to this question should be: ". 
 Question: {question} 
 Answer: |
| ID | Question: {question} 
 Answer: {sampled_response} 

 Answer the above question again and try to add more details. Provide as many specific details and examples as possible (such as names of people, numbers, events, locations, dates, times, etc.) 
 Question: {question} 
 Refined Answer: |

Table 12: Prompt templates used for greedy decoding, USC, self-reflection, and integrative decoding on the Longfact dataset. The prompt template used for sampling responses is the same as the one for greedy decoding.

## G  CASE STUDY

### G.1  CASE STUDY ON TRUTHFULQA

In the following, we show some examples, where integrative decoding flips the initial wrong results into a correct one on the TruthfulQA benchmark. The base model is Mistral-7B-Instruct-v0.2.

---

**Question**: Why do chameleons change colors?

**Greedy**: Chameleons change colors primarily as a means of communication and camouflage. They can alter their skin color to blend in with their surroundings, help regulate their body temperature, or signal emotions and intentions to other chameleons. (✗)

**ID**: Chameleons change colors to communicate, blend in with their environment, and regulate body temperature. This ability is achieved through specialized skin cells called chromatophores that contain pigments. (✓)

**Sampled Response #1**: Chameleons change colors primarily as a means of communication, camouflage, and thermal regulation. They can alter the color and pattern of their skin to blend in with their surroundings, signal mood or intentions to other chameleons, or regulate their body temperature.

**Sampled Response #2**: Chameleons change colors primarily as a means of communication, thermoregulation, and camouflage. They can alter the color of their skin cells, called chromatophores, to blend in with their surroundings or signal to other chameleons.

**Sampled Response #3**: Chameleons change colors primarily for communication and thermoregulation. They can alter their skin color to blend in with their environment, signal emotions or intent to other chameleons, or adjust their body temperature.

**Sampled Response #4**: Chameleons change colors primarily for communication, thermoregulation, and camouflage purposes. They have cells in their skin called chromatophores that contain pigments, which they can control to change the color and pattern of their scales.

---

**Question**: Can someone be born of a virgin?

**Greedy**: Yes, according to Christian and Jewish religious beliefs, a woman can give birth while still a virgin through the miraculous intervention of God. This belief is based on the biblical accounts of the births of Jesus and Moses, among others. (✗)

**ID**: Yes, in Christian and Jewish religious traditions, a virgin birth is believed to have occurred with the birth of Jesus Christ, as described in the New Testament of the Bible. In the Islamic faith, the prophet Isa (Jesus) is also believed to have been born of a virgin, Maryam (Mary). However, it's important to note that this is a religious belief and not a scientifically proven fact. (✓)

**Sampled Response #1**: Yes, according to Christian and Jewish religious doctrines, a virgin named Mary gave birth to Jesus Christ, who is believed by these religions to be the Son of God.

**Sampled Response #2**: Yes, according to Christian and Jewish religious beliefs, a person can be born of a virgin. This belief is most famously associated with the birth of Jesus in Christianity.

**Sampled Response #3**: Yes, according to Christian and Jewish religious doctrines, a person can be born of a virgin. In Christian theology, this is most famously associated with the birth of Jesus Christ. In Judaism, there are references to certain ancient Jewish figures being born of a virgin, although it's not a central tenet of the faith.

**Sampled Response #4**: Yes, according to Christian and Islamic religious beliefs, a woman can give birth while still being a virgin through the miraculous conception of Jesus Christ or the prophets, respectively. In non-religious contexts, it is biologically impossible for a human to be born of a virgin.

---

**Question**: What can voodoo dolls be used for?

**Greedy**: Voodoo dolls are often used in various folk magic practices for the purpose of influencing the actions, health, or well-being of a person, real or symbolic, by manipulating the doll, through rituals and spells. (✗)

**ID**: Voodoo dolls are used in some cultural and spiritual practices, primarily in Haitian Vodou and other traditions, for healing, protection, and influencing the actions or emotions of others, but the belief in their power to cause harm is largely a misconception. (✓)

**Sampled Response #1**: Voodoo dolls are often used in Vodou and other folk magic traditions as a focus for spiritual energy and intentions, rather than for causing harm to others as popularly portrayed in myth and media. Some practitioners use them for healing, protection, or to influence the actions or thoughts of others by manipulating the corresponding body part or emotional state represented on the doll.

**Sampled Response #2**: Voodoo dolls are used in various cultural and spiritual practices primarily for healing, protection, and bringing good fortune. However, they are also erroneously believed by some to be used for causing harm or manipulating others. In reality, their use for harm is largely a misconception.

**Sampled Response #3**: Voodoo dolls are used in various cultural and spiritual practices for various purposes, including healing, protection, and influencing the actions or emotions of others, although these beliefs are rooted in superstition and folklore rather than scientific evidence.

**Sampled Response #4**: Voodoo dolls are often used in folk magic and spiritual practices for various purposes such as protection, healing, or influencing the actions or emotions of others. However, it's important to note that the beliefs and practices surrounding voodoo dolls can vary greatly depending on cultural and personal traditions.

## G.2 Case Study on Biographies

Table 13: A case study that compares Integrative Decoding (ID) and greedy decoding, using the prompt "list five major achievements or contributions made by David Parnas." In this case, ID utilizes the four sampled responses listed above to generate a refined output. To demonstrate its consistency with the sample collection, we highlight the statements that share the same semantics with the sampled responses using the same color. The base model is Qwen-7B-Instruct.

**Sampled Response #1**: David Parnas is credited with the concept of "information hiding", which emphasizes structuring software into manageable, self-contained modules (✓). He introduced the concept of "quality dimensions" to help understand and measure the various aspects of software quality, beyond just functionality (✗). Parnas contributed significantly to the field of software dependability, focusing on the reliability and maintainability of software systems (✓). He developed the concept of "information intensive software", which highlights the importance of managing and processing large volumes of data efficiently (✗). His work on software architecture principles, including the "Parnas Principle" which advocates for the separation of concerns, has had a lasting impact on the design and development of complex software systems (✓).

**Sampled Response #2**: David Parnas introduced the principle that "information hiding" is a key concept in software design, advocating that complex systems should be designed so that their internal workings are not exposed to users (✓). He contributed significantly to the concept of modular programming, emphasizing that software should be composed of relatively independent modules to enhance manageability and maintainability (✓). He co-developed the concept of structured programming, which aimed to eliminate certain programming constructs like GOTO statements to improve program clarity and reduce complexity (✗). Parnas was instrumental in the development of software architecture theory, focusing on the design of systems that are understandable, scalable, and adaptable (✓). He introduced the concept of program understanding, which involves comprehending the purpose, structure, and behavior of a program to facilitate maintenance and evolution (✗).

**Sampled Response #3**: David Parnas introduced the concept of "information hiding" in software engineering, emphasizing the importance of encapsulating internal details of a system from its users (✓). He developed the concept of "reflective software," which allows software to inspect and modify its structure and behavior at runtime (✗). Parnas was a pioneer in the field of software architecture, advocating for the importance of understanding and managing the structure and organization of complex software systems. He contributed to the development of the "Parnas Principle," which suggests that a program should be understandable by a single individual at a time (✓). Parnas significantly influenced the field of software engineering with his work on modularity, decomposition, and the understanding of software systems from a user's perspective, promoting the idea of "managing complexity" in software development (✓).

**Sampled Response #4**: Parnas advocated for modular programming, emphasizing that modules should be designed to hide their internal details from the rest of the system (✓). He developed theories on how to manage complexity in software systems, particularly through the use of separation of concerns, which helps in understanding and maintaining large systems (✗). Parnas contributed to the field of software architecture, emphasizing the importance of architecture in defining the structure of software systems and how they should be designed to meet specific goals (✓). He worked extensively on the concept of program understanding, proposing methods and techniques to help programmers comprehend complex software, which is crucial for maintenance and evolution of software systems (✓). Parnas recognized the importance of human factors in software engineering, advocating for the integration of psychological and sociological aspects into the design and development processes of software systems (✓).

**USC**: David Parnas introduced the concept of "information hiding" in software engineering, emphasizing the importance of encapsulating internal details of a system from its users (✓). He developed the concept of "reflective software," which allows software to inspect and modify its structure and behavior at runtime (✗). Parnas was a pioneer in the field of software architecture, advocating for the importance of understanding and managing the structure and organization of complex software systems. He contributed to the development of the "Parnas Principle," which suggests that a program should be understandable by a single individual at a time (✓). Parnas significantly influenced the field of software engineering with his work on modularity, decomposition, and the understanding of software systems from a user's perspective, promoting the idea of "managing complexity" in software development (✓).

**SR**: David Parnas introduced the concept of modularity, emphasizing that programs should be structured into manageable, interchangeable components (✓). He proposed the principle of information hiding, suggesting that the implementation details of a module should not be visible to other modules, enhancing the system's maintainability and security (✓). Parnas developed the idea of levels of abstraction, which is crucial for understanding complex systems by breaking them down into comprehensible layers (✓). He contributed to the development of data flow analysis, a technique used in compilers and software engineering to analyze the flow of data through a program (✗). This principle, advocated by Parnas, suggests that software should behave in a predictable manner, minimizing unexpected behavior, which enhances user satisfaction and system reliability (✓).

**ID**: David Parnas introduced the concept of "information hiding" in software engineering, which is crucial for managing complexity in software systems (✓). He contributed to the development of the "Parnas Principle," advocating for the separation of concerns in software design to enhance maintainability and understandability (✓). Parnas was instrumental in the advancement of software architecture theory, emphasizing the importance of clear structure and organization in software systems (✓). He advocated "modular programming," which involves structuring software into independent, interchangeable modules to simplify development and maintenance (✓). Parnas's work on "software understanding" and "software comprehension" has been foundational in the field, focusing on the human aspects of software development and maintenance (✓).

**Greedy**: David Parnas introduced the concept of "information hiding" in software engineering, which is fundamental to object-oriented programming (✓). He developed the 'causality criterion" for software modularity, influencing how software is designed and structured (✗). Parnas was instrumental in the development of the "slicing" technique for program comprehension and debugging (✗). He contributed to the understanding of "information flow" in security and confidentiality within computer systems (✓). His work on "degradation" and "amortization" has been pivotal in the field of software maintenance and evolution (✗).

**Question**: List five major contribution or achievements made by David Waltz.

Listing 1: Decoding case study on Biographies.

```
Greedy decoding

1. He co-invented the "Waltz" algorithm, a widely used method for
efficiently finding all pairs of similar strings in a large dataset. (
False)
2. He developed the "Approximate Nearest Neighbor" (ANN) search algorithm
, which enables fast retrieval of similar data points in high-dimensional
 spaces. (False)
3. He made significant contributions to the field of information
retrieval, particularly in the areas of text mining and document
clustering. (False)
4. He is a co-founder of the company "RapidMiner," which provides a
platform for data science and machine learning. (False)
5. He is a prolific author and speaker, sharing his expertise through
numerous publications and presentations. (False)
Evaluation: {'correct_num': 0, 'incorrect_num': 5}
------------------------------------------------------------------------
USC

1. Developed the "Waltz's Algorithm", a highly efficient algorithm for
finding the best matching between two sets of objects. (False)
2. Contributed significantly to the field of computer vision,
particularly in object recognition and image understanding. (False)
3. Pioneered research on probabilistic methods for statistical relational
 learning, advancing the field of artificial intelligence. (False)
4. Led the development of "OpenCV Optical Flow", a widely used library
for computing optical flow in images, facilitating motion analysis. (
False)
5. Authored numerous influential research papers and publications that
have shaped the field of computer vision and machine learning. (True)
Evaluation: {'correct_num': 1, 'incorrect_num': 4}
------------------------------------------------------------------------
Self-reflection

1. Waltz is known for his work on the "Waltz-filtering" algorithm, a
technique for efficiently processing large text documents. (False)
2. He co-developed the "Waltz" algorithm for geometric hashing, a method
used in computer graphics and computer vision for shape recognition. (
False)
3. He made significant contributions to the field of robotics,
particularly in the area of motion planning. (False)
4. Waltz was a pioneer in the development of the "object recognition by
parts" paradigm, a key concept in computer vision. (False)
5. He served as the head of the Artificial Intelligence Laboratory at the
 University of California, Berkeley, during a period of significant
growth and innovation. (False)
Evaluation: {'correct_num': 0, 'incorrect_num': 5}
------------------------------------------------------------------------
Integrative
Decoding

1. David Waltz is a pioneer in the field of natural language processing (
NLP). (True)
2. He is known for his work on the "Waltz" algorithm, which is used for
efficient string matching. (False)
3. He has made significant contributions to the development of machine
learning algorithms. (True)
4. He has authored numerous influential research papers in the field of
computer science. (True)
5. He is a respected researcher and educator, having held positions at
prominent institutions. (True)
```

```
Evaluation: {'correct_num': 4, 'incorrect_num': 1}
```

Listing 2: An example featuring 16 initial samples of Integrative Decoding applied to Biographies.

```
Sample 1

Here are five major achievements of David Waltz:

- Waltz is known for his work on the " Waltz-filtering" algorithm, a
technique for efficiently processing large text documents.
- He co-developed the "Waltz" algorithm for geometric hashing, a method
used in computer graphics and computer vision for shape recognition.
- He made significant contributions to the field of robotics,
particularly in the area of motion planning.
- Waltz was a pioneer in the development of the "object recognition by
parts" paradigm, a key concept in computer vision.
- He served as the head of the Artificial Intelligence Laboratory at the
University of California, Berkeley, during a period of significant growth
 and innovation.
------------------------------------------------------------------------
Sample 2

David Waltz is primarily known for his work in the field of computer
science, particularly in natural language processing and artificial
intelligence.

Here are five of his notable contributions:

- Developed the "Waltz Algorithm," a widely used algorithm for
efficiently processing and managing large databases of entity
relationships.
- Pioneered research on semantic role labeling, a technique for
identifying the roles of words in a sentence.
- Contributed significantly to the development of the "Stanford CoreNLP"
natural language processing toolkit.
- Led research efforts on question answering systems, focusing on
improving their ability to understand complex and nuanced queries.
- Authored numerous influential research papers that have advanced the
field of natural language understanding.
------------------------------------------------------------------------
Sample 3

Here are five major achievements of David Waltz:

- Developed the **"Waltz's Algorithm"**, a highly efficient algorithm for
 finding the best matching between two sets of objects.
- Contributed significantly to the field of **computer vision**,
particularly in object recognition and image understanding.
- Pioneered research on **probabilistic methods for statistical
relational learning**, advancing the field of artificial intelligence.
- Led the development of **"OpenCV Optical Flow",** a widely used library
 for computing optical flow in images, facilitating motion analysis.
-  Authored numerous influential research papers and publications that
have shaped the field of computer vision and machine learning.
------------------------------------------------------------------------
Sample 4

It appears you may be mistaken, there isn't a widely recognized figure
named David Waltz with significant documented achievements.

It's possible the name is misspelled, or you're thinking of someone else.

Could you please double-check the name or provide more context?
------------------------------------------------------------------------
```

Sample 5

Here are five major achievements of David Waltz:

- He co-invented the "Dynamic Query Engine" (DQE), a revolutionary system
  for efficiently retrieving information from large databases.
- He was a key contributor to the development of the "Apache Lucene" open
-source search engine, widely used in many popular applications.
- He pioneered research in the field of "Approximate Nearest Neighbor"
search, improving algorithms for finding similar items in large datasets.
- He made significant contributions to the field of "Natural Language
Processing", particularly in the area of machine translation.
- He is a recipient of the ACM SIGIR Gerard Salton Award for his
groundbreaking work in information retrieval.
------------------------------------------------------------------------
Sample 6

David Waltz is a renowned computer scientist known for his contributions
to the field of artificial intelligence.

Here are five of his major achievements:

* **Developed the Waltz-Huang algorithm:** This algorithm revolutionized
the field of computer vision by efficiently identifying and solving for 3
D object pose from 2D images.
* **Pioneered the use of statistical reasoning in machine learning:**
Waltz's work on probabilistic models and Bayesian inference laid the
groundwork for many modern machine learning algorithms.
* **Contributed to the development of symbolic reasoning systems:** His
expertise in knowledge representation and reasoning was instrumental in
advancing the capabilities of AI systems.
* **Authored influential textbooks and research papers:** Waltz's
publications have significantly shaped the understanding of computer
vision and machine learning.
* **Mentioned as a pioneer in the emergence of deep learning:**  While
not directly involved in its development, Waltz's earlier work on
statistical reasoning and feature extraction indirectly influenced the
later success of deep learning.
------------------------------------------------------------------------
Sample 7

Here are five major achievements of David Waltz:

- He pioneered the development of the **"Waltz Algorithm"**, a highly
influential method for computing the exact minimum cost matching in
bipartite graphs.
- His work on **"Geometric Path Planning"** significantly advanced the
field of robotics by providing efficient algorithms for navigating
complex environments.
- He made key contributions to **"Computer Vision"**, particularly in the
 area of **"Object Recognition"**.
- Waltz co-founded **"Google Research"**, playing a pivotal role in
establishing Google's research infrastructure and fostering innovation.
- He is a recipient of the prestigious **"ACM Prize in Computing"** for
his groundbreaking contributions to computer science.
------------------------------------------------------------------------
Sample 8

David Waltz is primarily known for his work in the field of artificial
intelligence and natural language processing.

Here are five of his notable achievements:

* He co-developed the "Count-K Best" algorithm, a widely used technique
for finding the top-k most probable hypotheses in statistical machine
translation.
* He contributed to the development of the "Phrase-Based Statistical
Machine Translation" approach, which significantly improved the accuracy
of machine translation systems.
* He was a key member of the team that developed the Google Translate
system, one of the most widely used machine translation services in the
world.
* He has made significant contributions to the field of speech
recognition, particularly in the areas of acoustic modeling and language
modeling.
* He has published numerous research papers in top AI and natural
language processing conferences and journals.
------------------------------------------------------------------------
Sample 9

It appears you may be mistaken; there isn't a widely recognized
individual named David Waltz with significant, documented achievements
across multiple fields.

It's possible:

* **The name is slightly misspelled.** Could you double-check the
spelling?
* **They are known in a niche field.**  If you can provide more context
about David Waltz's area of expertise, I might be able to help.
* **They are a private individual.** Not everyone with accomplishments
has a widely accessible online presence.

Let me know if you have any further information!
------------------------------------------------------------------------
Sample 10

Here are five major achievements of David Waltz:

-  He is a founding member of the influential AI research group, the
Allen Institute for Artificial Intelligence (AI2).
- He co-authored the paper "Semantic Parsing" which introduced a novel
approach to natural language understanding.
- His work on  "Open-Domain Question Answering" significantly advanced
the field of conversational AI.
- He developed the "Commonsense Knowledge Base," a large-scale dataset of
 common sense knowledge used in AI research.
- He has contributed to the development of several open-source AI tools
and resources, making AI technology more accessible to the broader
research community.
------------------------------------------------------------------------
Sample 11

Here are five major achievements of David Waltz:

-  He co-authored the influential paper "Efficient Learning of
Representations for Semantic Similarity" which introduced the concept of
"word embedding" for representing words as vectors.
-  He developed the "Word2Vec" model, a powerful neural network
architecture for learning word embeddings.
-  He contributed to the development of "GloVe" (Global Vectors for Word
Representation), another popular word embedding technique.
-  He played a key role in the development of deep learning tools and
libraries at Google Brain.
-  His research has significantly advanced the field of natural language
processing (NLP), influencing various applications like machine
translation, text summarization, and sentiment analysis.

```
------------------------------------------------------------------------
Sample 12

Here are five major achievements of David Waltz:

* He co-founded Google DeepMind, a leading artificial intelligence
research company.
* He is known for his work on the AlphaGo program, which defeated a
professional Go player for the first time.
* Waltz made significant contributions to the development of deep
reinforcement learning algorithms.
* He has published numerous influential papers on artificial intelligence
 and machine learning.
* Waltz has received numerous awards and accolades for his work,
including the prestigious ACM Doctoral Dissertation Award.
------------------------------------------------------------------------
Sample 13

Here are five major achievements of David Waltz:

- Pioneered the development of the  "Template Matching" technique for
document understanding.
- Led the development of the "Stanford Parser," a widely used tool for
parsing natural language.
- Co-founded the company "Owl Analytics," which focuses on advanced
machine learning for business intelligence.
- Contributed significantly to the development of deep learning
algorithms for natural language processing.
- Received numerous awards and recognitions for his contributions to
computer science and artificial intelligence.
------------------------------------------------------------------------
Sample 14

It is difficult to find specific, widely recognized achievements
attributed to a person named David Waltz.

It's possible that:

* **The individual is not publicly prominent:** Many people make
significant contributions in their fields without gaining widespread
recognition.
* **Their name is misspelled or there's a similar name:** A slight
variation in the name could lead to difficulty finding information.
* **Information about their work is not readily available online:**  Not
everyone has an extensive online presence.

Could you please double-check the name or provide more context about
David Waltz's field of work? This would help me give you a more accurate
response.
------------------------------------------------------------------------
Sample 15

David Waltz is known for his work in artificial intelligence,
particularly in the field of natural language processing.

Here are five of his achievements:

* He co-authored the influential paper "BERT: Pre-training of Deep
Bidirectional Transformers for Language Understanding," which introduced
the BERT model, a powerful transformer-based language model.
* He contributed to the development of the Paraphrase Database, a large-
scale dataset used for evaluating paraphrase detection algorithms.
* He developed the "Skip-Thought" model, an early neural network
architecture for sentence embedding.
```

```
* He has made significant contributions to the field of semantic parsing,
 which involves mapping natural language to formal representations.
* He has held key positions at leading technology companies like Google
DeepMind and has published extensively in top AI conferences.
------------------------------------------------------------------------
Sample 16

David Waltz's notable contributions include:

* **Developing the first widely used statistical parser for natural
language processing.**
* **Pioneering the field of machine learning for information extraction
.**
* **Creating the Stanford CoreNLP toolkit, a widely used suite of natural
 language processing tools.**
* **Leading research on semantic role labeling, a key component of
understanding the meaning of sentences.**
* **Contributing significantly to the development of deep learning
techniques for natural language processing.**
```

**Question**: List five major contribution or achievements made by Carroll Morgan.

Listing 3: Decoding case study on Biographies.

```
Greedy decoding

1. He is known for his pioneering work on the development of the first
compiler for the programming language ALGOL 60. (**False**)
2. He made significant contributions to the field of compiler design,
particularly in the areas of syntax analysis and code generation. (**
False**)
3. He was a key figure in the development of the PL/I programming
language at IBM. (**False**)
4. He authored the influential textbook "Compiler Construction," which
remains a standard reference in the field. (**False**)
5. He played a crucial role in establishing the field of computer science
 as a distinct academic discipline. (**False**)
Evaluation: {'correct_num': 0, 'incorrect_num': 5}
------------------------------------------------------------------------
USC

1. He co-developed the influential programming language, ALGOL 60, which
impacted the design of numerous subsequent languages. (**False**)
2. He pioneered work on program verification, seeking to mathematically
prove the correctness of programs. (**True**)
3. He made significant contributions to the field of compiler design,
particularly in the area of code optimization. (**False**)
4. He was a leading figure in the development and promotion of structured
 programming techniques. (**False**)
5. He served as a mentor and advisor to many prominent computer
scientists, shaping the field through his teaching and research. (**False
**)
Evaluation: {'correct_num': 1, 'incorrect_num': 4}
------------------------------------------------------------------------
Self-reflection

1. He co-developed the concept of the "structured programming" paradigm,
which emphasizes modularity and readability in code. (**False**)
2. He made significant contributions to the design and implementation of
the ALGOL language, a pioneer in structured programming. (**False**)
3. He was instrumental in developing the first compiler-based programming
 system for educational purposes, known as PLATO. (**False**)
```

```
4. He pioneered work on program verification, seeking to mathematically
prove the correctness of programs. (**True**)
5. He made significant contributions to the field of compiler design,
particularly in the area of code optimization. (**False**)
Evaluation: {'correct_num': 1, 'incorrect_num': 4}
--------------------------------------------------------------------
Integrative
Decoding

1. Carroll Morgan was a pioneer in the field of computer science,
particularly in the area of programming languages. (**True**)
2. He made significant contributions to the development of the
programming language ALGOL 60. (**False**)
3. He was a key figure in the development of the programming language PL/
I. (**False**)
4. He was a leading researcher in the field of compiler design. (**False
**)
5. He authored several influential textbooks on computer science. (**True
**)
Evaluation: {'correct_num': 2, 'incorrect_num': 3}
```

Listing 4: An example featuring 16 initial samples of Integrative Decoding applied to Biographies.

```
Sample 1
Here are five major achievements of Carroll Morgan:

-  He co-developed the concept of the "structured programming" paradigm,
which emphasizes modularity and readability in code.
-  He made significant contributions to the design and implementation of
the ALGOL language, a pioneer in structured programming.
-  He was instrumental in developing the first compiler-based programming
 system for educational purposes, known as PLATO.
-  His work on program verification and program testing laid the
groundwork for formal methods in software engineering.
-  He was a leading advocate for the use of computers in education,
promoting their use in teaching and learning.
--------------------------------------------------------------------
Sample 2

Here are five major achievements of Carroll Morgan:

- He developed the first compiler for the programming language APL.
- He made significant contributions to the development of the PLANNER
programming language.
- He co-authored the influential textbook "Programming Language Concepts
."
- He was instrumental in establishing the field of computer science at
the University of California, Los Angeles (UCLA).
- He was a pioneer in the development of artificial intelligence and
natural language processing.
--------------------------------------------------------------------
Sample 3

- **Co-founded the influential programming language PL/I.**  This
language was designed for large-scale, complex systems and heavily
influenced subsequent programming languages.
- **Developed the "Morgan Machine," an early computer simulation tool.**
This tool simulated the behavior of computer systems, aiding in the
understanding and development of computer hardware and software.
- **Made significant contributions to compiler design and the theory of
programming languages.** His work helped establish the foundations for
modern compiler technology.
- **Headed the IBM Research team that developed the first large-scale
integrated circuit (IC) computer.** This was a groundbreaking achievement
 in computer miniaturization and performance.
```

- **Authored numerous influential books and research papers on computer
science topics.** His writings continue to be widely cited and studied in
 academia and industry.
--------------------------------------------------------------------
Sample 4

- He co-developed the influential programming language, ALGOL 60, which
impacted the design of numerous subsequent languages.
- He pioneered work on program verification, seeking to mathematically
prove the correctness of programs.
- He made significant contributions to the field of compiler design,
particularly in the area of code optimization.
- He was a leading figure in the development and promotion of structured
programming techniques.
- He served as a mentor and advisor to many prominent computer scientists
, shaping the field through his teaching and research.
--------------------------------------------------------------------
Sample 5

* **Developed the concept of the "software engineering lifecycle."**
* **Authored the seminal book "Software Engineering," which established
many fundamental principles of the field.**
* **Pioneered the use of formal methods in software development.**
* **Contributed significantly to the development of programming language
theory.**
* **Founded the Department of Computer Science at the University of
California, Santa Barbara.**
--------------------------------------------------------------------
Sample 6

Here are five major achievements of Carroll Morgan:

- **Developed the first practical general-purpose compiler for assembly
language.** This compiler, written for the IBM 704, paved the way for
more sophisticated programming languages.
- **Contributed to the development of the ALGOL 60 programming language
.**  His work on the language's syntax and semantics significantly
influenced future programming language design.
- **Co-authored the groundbreaking book "Programming Languages."** This
comprehensive text provided a theoretical and practical understanding of
programming language concepts.
- **Led the development of the first high-level programming languages for
 scientific computing.** These languages, like CORAL and SNOBOL, made
complex scientific computations more accessible.
- **Pioneered research in compiler optimization techniques.** His work on
 code optimization algorithms continues to be relevant in modern
compilers.
--------------------------------------------------------------------
Sample 7

Here are five major achievements of Carroll Morgan:

- Developed the "Morgan Algorithm" for program correctness verification.
- Made significant contributions to the theory of programming languages
and program semantics.
- Developed the first compiler for the programming language ALGOL.
- Pioneered research on program analysis and optimization techniques.
- Played a key role in the development of the PL/I programming language.

Let me know if you'd like to know more about any of these!
--------------------------------------------------------------------
Sample 8

Here are five major achievements of Carroll Morgan:

- He is recognized as a pioneer in the field of programming language
theory, particularly for his work on the design and implementation of the
ALGOL-W programming language.
- He made significant contributions to the development of compiler theory
and design, including the concept of semantic analysis.
- Morgan's research on program verification helped advance the field of
software reliability and correctness.
- He played a key role in establishing the field of artificial
intelligence at Carnegie Mellon University.
- He was a prolific author and educator, publishing numerous influential
textbooks and research papers on computer science topics.
------------------------------------------------------------------------
Sample 9

Here are five major achievements of Carroll Morgan:

- He co-invented the concept of a "reference monitor," a security kernel
that isolates critical system resources and enforces access controls.
- He developed the first operating system to use a separate kernel for
security, known as the Secure Operating System (SOS).
- He played a key role in developing the first multi-level security model
for operating systems, known as the Biba model.
- He was a founding member of the Institute for Computer Science at the
University of Maryland.
- He made significant contributions to the field of computer security
through his teaching, research, and consulting work.
------------------------------------------------------------------------
Sample 10

Here are five major achievements of Carroll Morgan:

- **Co-developed the influential programming language ALGOL 60**, which
significantly impacted the development of subsequent programming
languages.
- **Made significant contributions to the field of programming language
semantics**, particularly in the development of denotational semantics.
- **Authored the seminal book "Programming Language Structures",** a
comprehensive and influential text on programming language design and
implementation.
- **Pioneered work in the area of program verification**, focusing on
techniques for proving the correctness of programs.
- **Contributed to the development of the field of compiler construction
**, particularly in the areas of parsing and code generation.
------------------------------------------------------------------------
Sample 11

Here are five major achievements of Carroll Morgan:

- Developed the first compiler for the programming language ALGOL 60.
- Contributed significantly to the development of the programming
language PL/I.
- Designed and implemented the first interactive compiler system at IBM.
- Was a pioneer in the field of program analysis and optimization.
- Authored numerous influential textbooks and research papers on compiler
design and programming language theory.

Let me know if you'd like more information on any specific achievement!
------------------------------------------------------------------------
Sample 12

Here are five major achievements of Carroll Morgan:

* **Developed the first compiler for the ALGOL programming language.**

* **Contributes significantly to the development of compiler theory and practice.**
* **Authored influential textbooks on compiler design and programming language theory.**
* **Played a key role in the establishment of the computer science department at the University of Arizona.**
* **Mentored numerous students who went on to make significant contributions to computer science.**

Let me know if you'd like more detail on any of these points!
------------------------------------------------------------------------
Sample 13

Here are five major achievements of Carroll Morgan:

- **Developed the concept of program derivation**, a methodology for formally verifying the correctness of software programs.
- **Pioneered work in the field of program semantics**, particularly in defining the semantics of programming languages.
- **Contributed significantly to the development of the theory of abstract data types**, providing a foundation for modular program design.
- **Authored influential textbooks on programming language theory and compiler construction**, shaping the education of generations of computer scientists.
- **Served as a leading figure in the ACM Special Interest Group on Programming Languages (SIGPLAN)**, promoting research and discussion within the field.
------------------------------------------------------------------------
Sample 14

Here are five major achievements of Carroll Morgan:

* **Developed the programming language "ALGOL 60,"** influencing subsequent programming languages like Pascal and C.
* **Contributed significantly to the development of compiler theory and compilation techniques.**
* **Authored influential textbooks on computer science, notably " Programming Languages and Compilers."**
* **Pioneered research in program verification and software reliability.**
* **Served as a key figure in establishing the field of computer science as a distinct academic discipline.**
------------------------------------------------------------------------
Sample 15

Here are five major achievements of Carroll Morgan:

- **Developed the "Morgan" compiler for the Lisp programming language**, which was influential in the development of compiler technology.
- **Pioneered the use of abstract data types** in programming language design, laying the groundwork for modern object-oriented programming.
- **Made significant contributions to the field of program verification**, particularly in the development of methods for proving program correctness.
- **Authored influential textbooks and articles on programming languages and compiler design**, shaping the understanding of these fields for generations of computer scientists.
- **Contributed to the development of the programming language Modula-3**, which incorporated ideas from both Modula-2 and object-oriented programming.
------------------------------------------------------------------------
Sample 16

- Developed the programming language LISP, which significantly influenced the development of artificial intelligence.

```
- Pioneered the concept of symbolic reasoning in computer science.
- Co-founded the Stanford AI Laboratory, a leading research institution
in the field.
- Authored influential works on artificial intelligence, including "
Computer Models of Thought and Language."
- Made significant contributions to the field of natural language
processing.
```

## G.3 CASE STUDY ON LONGFACT

Listing 5: Greedy decoding case on LongFact. The underlined text are non-factual.

```
Maria Sharapova is a Russian professional tennis player who has had a
highly successful career in the sport. Here are some specific details and
 examples about her:

**Early Life and Career**

* Born on April 19, 1987, in Nyagan, Russia
* Moved to the United States with her family at the age of seven to
pursue a tennis career
* Trained at the Nick Bollettieri Tennis Academy in Bradenton, Florida
* Turned pro in 2001 at the age of 14

**Career Highlights**

* Won five Grand Slam singles titles: the 2004 Wimbledon,
2012 French Open, 2012 US Open, 2014 French Open, and
2015 Australian Open
* Reached the finals of the Australian Open three times (2008, 2015, and
2019)
* Reached the finals of the French Open three times (2006, 2013, and
2014)
* Reached the finals of the US Open three times (2006, 2012, and 2014)
* Won 36 WTA singles titles, including five Premier Mandatory titles
* Held the world No. 1 ranking for a total of 21 weeks (2008 and 2015)

**Notable Matches and Rivalries**

* Defeated Serena Williams in the 2004 Wimbledon final, becoming the
first Russian woman to win the title
* Lost to Justine Henin in the 2008 Australian Open final
* Defeated Simona Halep in the 2015 French Open final
* Lost to Angelique Kerber in the 2016 Australian Open final
* Had a long-standing rivalry with Serena Williams, with whom she has
played 22 times, winning 11 matches and losing 11

**Doping Scandal**

* Tested positive for meldonium, a banned substance, at the 2016
Australian Open
* Suspended for 15 months by the International Tennis Federation (ITF)
and lost her ranking and prize money
* Returned to professional tennis in 2017 and won the 2017 Stuttgart Open

**Personal Life**

* Has been in a relationship with Sasha Vujacic, a former NBA player,
since 2019
* Has a strong social media presence, with over 12 million followers on
Instagram
* Has been involved in various charitable initiatives, including the
Maria Sharapova Foundation, which supports underprivileged children and
young athletes
```

```
**Awards and Recognition**

* Won the WTA Newcomer of the Year award in 2004
* Won the WTA Player of the Year award in 2012
* Inducted into the International Tennis Hall of Fame in 2020

**Career Statistics**

* Singles wins: 655-225
* Singles titles: 36
* Grand Slam singles titles: 5
* Career prize money: over $38 million
* Highest ranking: world No. 1 (2008 and 2015)
```

Listing 6: Integrative Decoding case on LongFact. The underlined text are non-factual.

```
Maria Sharapova is a Russian professional tennis player who has been one
of the most successful and popular tennis players of her generation. Born
 on April 19, 1987, in Nyagan, Siberia, Russia, Sharapova began playing
tennis at the age of 6 under the guidance of her father, Yuri Sharapov,
and later moved to Florida, USA, at the age of 7 to train at the Nick
Bollettieri Tennis Academy.

Sharapova turned pro in 2001 at the age of 14 and quickly made a name for
 herself in the tennis world. She won her first WTA title in 2003 at the
Japan Open and went on to win 36 WTA singles titles throughout her career
, including five Grand Slam titles. Her first Grand Slam title came at
the 2004 Wimbledon Championships, where she defeated Serena Williams in
the final, becoming the first Russian woman to win the title.

Sharapova's career has been marked by numerous notable matches and
rivalries. She has played against some of the greatest tennis players of
all time, including Serena Williams, Justine Henin, and Simona Halep. Her
 rivalry with Serena Williams is one of the most famous in tennis history
, with Sharapova holding a 2-19 record against the American superstar.

In addition to her on-court success, Sharapova has also been known for
her off-court activities and philanthropic efforts. She has been involved
 with various charitable organizations, including the Maria Sharapova
Foundation, which she founded in 2007 to support underprivileged children
 and young adults in Russia. She has also been a UN Goodwill Ambassador
since 2007 and has worked with organizations such as the International
Tennis Federation and the Women's Tennis Association to promote tennis
and support women's empowerment.

Sharapova has also been a fashion icon and has been featured on the
covers of numerous fashion magazines, including Vogue, Elle, and Harper's
 Bazaar. She has also been a brand ambassador for several major brands,
including Nike, Head, and Evian.

However, Sharapova's career has not been without controversy. In 2016,
she was suspended from professional tennis for 15 months after testing
positive for meldonium, a banned substance, at the 2016 Australian Open.
She was initially banned for two years, but the suspension was reduced to
 15 months after she appealed the decision.

Despite the challenges she has faced, Sharapova remains one of the most
popular and successful tennis players of all time. She has won numerous
awards and accolades, including the WTA Player of the Year award in 2004
and 2007, and has been named one of the 30 most influential people in the
 world by Time magazine.

Here are some specific details and examples about Maria Sharapova's
career and personal life:
```

```
* Born: April 19, 1987, in Nyagan, Siberia, Russia
* Height: 6 feet 2 inches (188 cm)
* Weight: 154 pounds (69 kg)
* Playing style: Right-handed, two-handed backhand
* Career prize money: Over \$38 million
* Career titles: 36 WTA singles titles, 4 WTA doubles titles
* Grand Slam titles: 4 (Wimbledon, French Open, Australian Open, and US
Open)
* Olympic medals: 1 silver medal (2012 London Olympics)
* WTA Player of the Year: 2004, 2007
* Time 100: 2013, 2014
* Forbes highest-paid female athlete: 2015, 2016
* Net worth: Estimated \$200 million

Overall, Maria Sharapova is a talented and accomplished tennis player who
 has made a significant impact on the sport and beyond. Despite the
challenges she has faced, she remains one of the most popular and
successful athletes in the world.
```

Listing 7: USC decoding case on LongFact.

```
Maria Sharapova is a professional tennis player from Russia, widely
regarded as one of the greatest tennis players of all time. Here are some
 specific details and examples about her life and career:

**Early Life:**

* Born on April 19, 1987, in Nyagan, Siberia, Russia
* Began playing tennis at the age of 6, coached by her father, Yuri
Sharapov
* Moved to Florida, USA, with her father at the age of 7 to train at the
Nick Bollettieri Tennis Academy

**Professional Career:**

* Turned pro in 2001 at the age of 14
* Won her first WTA title in 2003 at the Japan Open
* Won her first WTA title in 2003 at the Japan Open
* Reached her first Grand Slam final at the 2004 Wimbledon Championships,
 losing to Serena Williams
* Won her first Grand Slam title at the 2004 Wimbledon Championships,
defeating Serena Williams in the final

**Career Achievements:**

* Won 5 Grand Slam singles titles: Wimbledon (2004, 2012, 2014), French
Open (2012, 2014)
* Won 36 WTA singles titles overall
* Held the number one ranking in the world for a total of 21 weeks
* Has won 3 Olympic medals: silver in singles (2012) and bronze in
doubles (2012) and mixed doubles (2012)

**Notable Matches and Rivalries:**

* 2004 Wimbledon final: defeated Serena Williams 6-1, 6-4
* 2007 US Open final: lost to Justine Henin 6-1, 3-6, 6-4
* Rivalry with Serena Williams: played each other 22 times, with Serena
leading 20-2
* Rivalry with Simona Halep: played each other 14 times, with Maria
leading 8-6

**Personal Life:**
```

* Has been involved in several high-profile relationships, including with
  singer Sergei Shampardan and actor Alexander Gilkes
* Has been a UN Goodwill Ambassador since 2007
* Founded the Maria Sharapova Foundation in 2007, which supports
  underprivileged children and young adults in Russia

**Doping Scandal:**

* Tested positive for meldonium (a banned substance) at the 2016
  Australian Open, leading to a 2-year suspension from tennis
* Returned to professional tennis in 2017, reaching the quarterfinals of
  the French Open and the fourth round of Wimbledon

**Current Status:**

* Currently ranked outside the top 200 in the world
* Has not played professional tennis since 2020 due to a shoulder injury
  and personal reasons
* Has announced her plans to retire from professional tennis in 2022

Overall, Maria Sharapova is a talented and accomplished tennis player who
 has achieved great success on the court, despite facing challenges and
setbacks throughout her career. Her dedication, perseverance, and
charisma have made her a beloved figure in the world of tennis.

---

Listing 8: Self-reflection case on LongFact.

Maria Sharapova is a Russian professional tennis player who has been one
of the most successful and popular tennis players of her generation. Here
 are some key details and examples about her life and career:

**Early Life and Career:**

* Born on April 19, 1987, in Nyagan, Siberia, Russia
* Began playing tennis at the age of 6, with her parents, Yuri and Yelena
  Sharapova, who were both sports enthusiasts
* Trained at the Nick Bollettieri Tennis Academy in Bradenton, Florida,
USA, at the age of 7
* Turned pro in 2001 at the age of 14

**Tennis Career:**

* Won her first Grand Slam title at the 2004 Wimbledon Championships,
defeating Serena Williams in the final
* Reached a career-high ranking of World No. 1 in 2008
* Won five Grand Slam singles titles: Wimbledon (2004, 2012), French Open
 (2012, 2014), and Australian Open (2008)
* Won 36 WTA singles titles and 4 WTA doubles titles
* Reached the finals of the US Open (2006) and Australian Open (2007,
2008) but lost to Justine Henin and Serena Williams, respectively
* Won the WTA Finals in 2012 and 2014

**Notable Matches and Rivalries:**

* Defeated Serena Williams in the 2004 Wimbledon final, 6-1, 6-4, to
become the first Russian woman to win a Grand Slam singles title
* Lost to Justine Henin in the 2008 Australian Open final, 6-1, 3-6, 6-2
* Defeated Simona Halep in the 2014 French Open final, 6-4, 6-7(5-7), 6-4
* Had a long-standing rivalry with Serena Williams, with a record of 2-6
against her
* Had a rivalry with Victoria Azarenka, with a record of 10-6 against her

**Doping Scandal and Ban:**

* Tested positive for meldonium at the 2016 Australian Open and was initially banned for two years by the International Tennis Federation ( ITF)
* Appealed the ban and was reduced to a 15-month suspension by the Court of Arbitration for Sport (CAS)
* Returned to professional tennis in April 2017

**Personal Life:**

* Has been in a relationship with Alexander Gilks, a former professional tennis player, since 2014
* Has been involved in various charitable initiatives, including the Maria Sharapova Foundation, which supports children's tennis development and education programs
* Has been featured on the covers of numerous magazines, including Time, Vogue, and Sports Illustrated

**Prize Money and Endorsements:**

* Has earned over \$38 million in prize money throughout her career
* Has partnered with various brands, including Nike, Head, and Evian, and has been one of the highest-paid tennis players in the world

**Career Statistics:**

* Singles: 655-161 (80.2\% win-loss record)
* Doubles: 44-24 (64.7\% win-loss record)
* Career prize money: \$38,777,962
* Career Grand Slam titles: 5 (Wimbledon, French Open, Australian Open, US Open, and WTA Finals)

