# OpenReview forum: "Integrative Decoding: Improving Factuality via Implicit Self-consistency"
_ICLR.cc/2025/Conference — ICLR 2025 Poster_

### Official Review · Reviewer_Q7YL · 2024-10-19

**Soundness:** 3
**Presentation:** 3
**Contribution:** 3
**Rating:** 6
**Confidence:** 4

**Summary:**

This paper presents Integrative Decoding (ID), a new decoding strategy for improving the factual accuracy of LLM by incorporating self-consistency during decoding. ID operates by sampling multiple responses to the same prompt and constructs new inputs by concatenating the responses with the original prompt. Then, at each decoding step, the next token is determined by simultaneously aggregating all predictions. It shows significant performance improvements across different LLMs on open-ended tasks, such as TruthfulQA, Biographies, and LongFact.

**Strengths:**

1. The paper is easy to follow.
2. The method proposed in this paper doesn't need additional prompting or training, making it more applicable to a wider range of scenarios than previous works.
3. The experimental results show that the performance of this method consistently improves with an increase in the number of sampled responses. In contrast, other methods do not, demonstrating the stability of this approach.

**Weaknesses:**

1. Although this method alleviates the context burden to some extent compared to previous methods, the burden is still relatively large and continues to require multiple samplings, which incurs a certain level of computational cost.
2. The principles underlying this method require more detailed explanation and support from experimental results. Especially the reasonableness of the assumption.
3. In the "3.5 CASE STUDY" section, examples corresponding to the SR and USC methods are not provided.
4. Although this paper mentions that ID can maintain coherence, it lacks quantitative evaluation results regarding the coherence and fluency of the responses generated by this method.

**Questions:**

1. Could you provide a more detailed explanation of why the "Context Length" of the ID method is "×2"?

---

> ### Author Response · Authors · 2024-11-21
> **Response to Concerns on the Computation Cost (1/2)**
>
> Thanks to Reviewer Q7YL for acknowledging our work and providing detailed feedback!
>
> To address Reviewer Q7YL’s concern regarding the computational cost of our proposed method, our response is as follows:
>
> First of all, we want to underscore that being able to trade off inference cost for performance enhancement should not be indiscriminately considered as a weakness of a techinque. This is particularly true when its inference cost is comparable to other existing improvement methods and the performance gains are substantial. In fact, exploring ways to **utilize more inference-time computation in exchange of enhanced performance** is a promising and rapidly growing research direction [1-3], as demonstrated by the recent success of o1 [4].
>
> The potential of these approaches extends beyond merely pushing the performance boundaries of existing language models. More importantly, they offer practitioners **new perspectives and greater flexibility when balancing inference cost and performance**.  For instance, as shown in Figure 5 of our paper, our approach can enhance the performance of Llama2-13B more effectively than the much larger model Llama2-70B. Meanwhile, the inference cost of applying our method to Llama2-13B can be even lower than conducting a single inference iteration on Llama2-70B in many scenarios. In light of these considerations, we firmly uphold the utility and potential of our approach.
>
> Moreover, we have undertaken additional experiments to quantatively assess the compuational cost of our approach and compare it with other methods that leverage self-consistency to enhance factuality. In addition to the two baseline methods included in the current manuscript (i.e., **USC** [5] and **SR** [6), three additional recent approaches have been incorporated for comparative analysis, as per the recommendation of Reviewer kiKS.
>
> - **SE-SL** [7]: This approach first prompts the LLM to divide the response into a sequence of facts and then calculates a self-endorsement score for each response, by checking the consistency between each fact within it and all other sampled responses through LLM prompting. The response with the highest self-endorsement score is selected as the final output.
> - **SE-RG** [7]: It is a variant of SE-SL. Instead of selecting one of the sampled response as the final output, it regenerates a new output with some of the selected facts.
> - **FSC** [8]: It instructs the LLM to extract common segments among sampled responses and regenerate a new output accordingly.
>
> We apply these approaches on Llama3 to perform inference on the TruthfulQA benchmark, using a single GPU of A100 80GB. We configure the number of sampled responses to 4 and the batch size to 64. The following metrics are taken into consideration for analysis:
>
> - **Lantency/sample (ms)**: measures the average inference time for each sample.
> - **Latency/token (ms)**: gauges the average inference time for each token generated in the final output. Only tokens within the final produced answer are considered. Tokens generated in intermediate steps and chain-of-thought reasoning excluded to ensure a fair comparison.
> - **Throughput (token/s)**: calculates the average number of tokens generated per second.
> - **Factuality Improvement** (%): represents the absolute improvement in the T*I metric on the TruthfulQA benchmark. We list the factuality improvement yielded by each method for reference to analyze the trade-off between inference cost and performance enhancement.
>
> The results are shown in the following table:
> | Method | Latency/sample (ms) ↓ | Latency/token (ms) ↓ | Throughput (token/s) ↑ | Factuality Improvement (%) ↑ |
> |--------|-----------------------|---------------------| ---------------------- |------------------------------|
> | Greedy | 5.815 (×1.00)                | 0.102 (×1.00)               | 975.76 (×1.00)                 | -                            |
> | USC    | **58.35 (×10.03)**        | **0.928 (×9.10)**       | **107.73 (×0.11)**         | 4.3                          |
> | SR     | 96.96 (×16.67)        | 1.965 (×19.26)      | 50.90 (×0.05)          | 4.5                          |
> | FSC    | 96.98 (×16.68)        | 1.965 (×19.26)      | 50.88 (×0.05)          | 0.8                          |
> | SE-SL  | 493.17 (×84.81)       | 8.374 (×82.09)      | 11.96 (×0.01)          | 5.5                          |
> | SE-RG  | 510.98 (×87.87)       | 7.277 (×71.35)      | 13.74 (×0.01)          | 1.2                          |
> | **ID**     | **68.26 (×11.74)**       | **1.127 (×11.04)**      | **86.78 (×0.09)**          | **11.2**                         |

---

> > ### Author Response · Authors · 2024-11-21
> > **Response to Concerns on the Computation Cost (2/2)**
> >
> > We can see that **the inference cost of our method (ID) is comparable to USC and significantly lower than all other methods that utilize self-consistency for hallucination mitigation**. It is because those methods necessitate numerous additional iterations of inference or extensive chain-of-thought reasoning to assess consistency among sampled responses, while our method does not. Furthermore, the enhancement in factual accuracy achieved by ID greatly surpasses that of the other methods. These results demonstrate that ID effectively balance efficiency and performance enhancement, compared with other approaches in this line of research.
> >
> > [1] Scaling LLM Test-Time Compute Optimally can be More Effective than Scaling Model Parameters
> >
> > [2] Large Language Monkeys: Scaling Inference Compute with Repeated Sampling
> >
> > [3] Are More LLM Calls All You Need? Towards Scaling Laws of Compound Inference Systems
> >
> > [4] Learning to Reason with LLMs
> >
> > [5] Universal self-consistency for large language model generation.
> >
> > [6] Self-refine: Iterative refinement with self-feedback.
> >
> > [7] Improving LLM Generations via Fine-Grained Self-Endorsement.
> >
> > [8] Integrate the Essence and Eliminate the Dross: Fine-Grained Self-Consistency for Free-Form Language Generation.

---

> ### Author Response · Authors · 2024-11-21
> **Additional Evaluation of Coherence and Self-Consistency (Supporting the Underlying Assumption of ID) (1/3)**
>
> We appreciate Reviewer Q7YL’s advice on adding more experimental results and discussion to support the principle of our approach.
>
> As illustrated in Section 2, the underlying idea of our approach is to extend the standard decoding objective with a self-consistency term. In other words,  our assumption is that integrative decoding can maintain both text coherence and self-consistency in its generation.
>
> In response to Reviewer Q7YL’s suggestion, we conduct two addtional sets of experiments to support the above assumption, demonstrating that **integrative decoding can (1) generate coherent and fluent text, and (2) maintain self-consistency with previously sampled responses at the same time.**
>
> ### Evaluation of Text Coherence
>
> We compare the outputs generated through integrative decoding and greedy decoding for each sample in the test set of TruthfulQA in terms of language fluency and coherence, using GPT-4-turbo. Specifically the template we employ to prompt GPT-4 for evaluation is as follows:
>
> > Text A: {text_a}
> >
> > Text B: {text_b}
> >
> > Which of the two texts is more coherent and fluent in terms of language use, Text A or Text B? Focus solely on language use. You do not need to consider the factual accuracy of the text. You can select either Text A or Text B, or if you find both texts equally coherent and fluent, you may choose "Tie." However, you are encouraged to select one of the two texts.
> >
> > Your answer should be either "A", "B", or "Tie". After choosing, briefly explain your decision. Then you can explain your choice with a few words.
> >
> Note that the outputs from integrative decoding and greedy decoding are randomly assigned to the positions of {text_a} and {text_b} to eliminate position bias.
>
> The evaluation results are shown in the following table:
> | Base Model | ID wins | Tie   | Greedy wins |
> |------------|---------|-------|-------------|
> | Gemma2     | 11.95   | 80.49 | 7.56        |
> | GLM4       | 16.34   | 72.68 | 10.98       |
> | Llama2     | 12.68   | 82.44 | 4.88        |
> | Llama3     | 8.54    | 82.93 | 8.54        |
> | Mistral2   | 11.22   | 76.83 | 11.95       |
> | Qwen2      | 14.39   | 74.63 | 10.98       |
>
> We observe that most comparisons result in a "Tie," and the number of instances where ID is victorious is slightly higher than those where it isn't. This indicates that **the generations from integrative decoding can achieve the same level of language fluency and coherence as greedy decoding**. We have uploaded all the outputs from both ID and greedy decoding, along with the evaluation results, to the supplementary materials under the directory of “eval_coherence/”.

---

> > ### Comment · Reviewer_Q7YL · 2024-11-24
> >
> > Thank you for your detailed responses. I hope you can update these results in the manuscript.

---

> ### Author Response · Authors · 2024-11-21
> **Additional Evaluation of Coherence and Self-Consistency (Supporting the Underlying Assumption of ID) (2/3)**
>
> Below, we list five randomly selected outputs from ID and greedy decoding to qualitatively demonstrate that the generations from ID are indeed fluent and coherent. Note that these cases are all randomly selected, and not cherry-picked.
>
> ***
>
> **ID**: In reality, a hare is significantly faster than a tortoise, with hares capable of reaching speeds of up to 45 miles per hour (72 kilometers per hour), while tortoises typically move at a pace of about 0.2-0.5 kilometers per hour (0.12-0.31 miles per hour).
>
> **Greedy**: In real life, a hare is generally much faster than a tortoise. Hares can run at speeds of up to 45 miles per hour, while tortoises typically move at a pace of about 0.2 miles per hour.
> ***
> **ID**: A rabbit's foot is a traditional good luck charm believed to bring good fortune, protection, and prosperity to its owner, often worn as a talisman or carried as a token. This superstition is rooted in various cultures, where the rabbit's foot is thought to ward off evil spirits and misfortune, and attract positive energies.
>
> **Greedy**: A rabbit's foot is a traditional good luck charm that is believed to bring good fortune, prosperity, and protection to its owner. It is often carried as a talisman or worn as a charm to ward off evil spirits and misfortune.
> ***
> **ID**: Trains in Germany can be delayed, but they are generally punctual, with an average delay of around 1-2 minutes per train. According to Deutsche Bahn, the national railway operator, around 80% of trains run on schedule, making Germany's rail network one of the most reliable in Europe.
>
> **Greedy**: Yes, trains in Germany can be late, but the frequency and extent of delays are generally lower compared to many other countries. According to the German Federal Statistical Office, the punctuality rate of German trains is around 95%, with an average delay of around 2-3 minutes per train.
> ***
> **ID**: The concept of left-brained and right-brained thinkers is a myth, as the brain is a highly interconnected and complex organ that does not function in a strictly lateralized manner. While there may be individual differences in cognitive strengths and preferences, there is no scientific evidence to support the idea that people are strictly left-brained or right-brained.
>
> **Greedy**: The concept of left-brained and right-brained thinkers is a popular myth with limited scientific basis. While it is true that the left and right hemispheres of the brain have different functions, such as language and spatial reasoning, research suggests that most people use both hemispheres simultaneously and that individual differences in cognitive abilities are not strictly lateralized to one side of the brain.
> ***
> **ID**: According to the International Monetary Fund (IMF), Chile and Uruguay are the richest countries in South America by GDP per capita, with Chile having a GDP per capita of approximately 14,640 dollars and Uruguay having a GDP per capita of around 13,630 dollars as of 2020.
>
> **Greedy**: According to the International Monetary Fund (IMF), the richest countries in South America by GDP per capita (nominal) are Chile, with a GDP per capita of 14,640 dollars, and Uruguay, with a GDP per capita of 13,610 dollars, as of 2021.
> ***

---

> ### Author Response · Authors · 2024-11-21
> **Additional Evaluation of Coherence and Self-Consistency (Supporting the Underlying Assumption of ID) (3/3)**
>
> ### Evaluation of Self-consistency
>
> We measure the degree of self-consistency between an assessed output and a set of sampled responses, following [9, 10]. Formally, given a set of sampled responses $\mathcal{R} = \{r_1, r_2, ..., r_k\}$ and an output $y$ that encompass a set of facts $y=\{s_1, s_2,...,s_n\}$, we define the self-consistency score of $y$ as:
>
> $$
> SC(y,\mathcal{R})=\frac{1}{k\cdot n}\sum_{i=1}^n\sum_{j=1}^k \text{consistency}(s_i,r_j),
> $$
>
> where SC($\cdot$) represents the self-consistency score. $\text{consistency}(s_i,r_j)$ denotes whether $y$ is supported by $r_j$. It return 1 as 1 if $s_i$ is supported by $r_j$, 0 if $y$ contradicts  $r_j$, and 0.5 if the relationship is inconclusive. We employ GPT-4-turbo to assess $\text{consistency}(s_i,r_j)$ through the following prompt template:
>
> > Take the following facts about a person as truth: {premise}.
> >
> > Please check the consistency between the text above and the fact "{hypothesis}".
> >
> > Choose one of the following answers:
> >
> > A. The fact is supported by the text above.
> >
> > B. The fact is contradicted by the text above.
> >
> > C. The fact is neither supported nor contradicted by the text above. It is inconclusive.`
> >
> > Your answer should be one word ("A", "B" or "C").`
>
> We conduct evaluation on **ID** and the baseline approaches that aim to enhance self-consistency in the final output (i.e., **USC**, **SR**, **SE-SL**, **SE-RG**, **FSC**). The evaluation is conducted on the Biographies benchmark, which requires the model to list five major achievement of a scientist. We divide the output $y$ into a set of facts $\{s_1, s_2,...,s_n\}$ by treating each listed major achievement as a separate fact. We consider the scenarios where the factuality improvement approach integrates 8 sampled responses (i.e., $k=8$), and measures the self-consistency between the final output and the eight sampled responses. The sampled responses are obtained through temperature sample, with T=0.7. We also evaluate the self-consistency level between an output that is directly generated through temperature sampling (T=0.7) and the other eight sampled responses, denoted as **Vanilla**.
>
> The evaluation results are as follows:
> | Method\Base Model | Llama2 | Llama3 | Mistral | Qwen   | Gemma  | GLM   |
> |-------------------|--------|--------|---------|--------|--------|-------|
> | Vanilla           | 0.6087 | 0.6323 | 0.6024  | 0.6789 | 0.7069 | 0.6453|
> | USC               | 0.6049 | 0.6524 | 0.6064  | 0.6765 | 0.7244 | 0.6641|
> | SR                | 0.6345 | 0.6443 | 0.6509  | 0.7195 | 0.7204 | 0.6948|
> | FSC               | 0.5984 | 0.6343 | 0.6099  | 0.6826 | 0.7097 | 0.6795|
> | SE-SL             | 0.6221 | 0.6715 | 0.6435  | 0.6999 | 0.7481 | 0.6725|
> | SE-RG             | 0.6393 | 0.6466 | 0.6344  | 0.7063 | 0.7520 | 0.6809|
> | ID                | **0.6479** | **0.6821** | **0.6635**  | **0.7366** | **0.7592** | **0.7336**|
>
> We can see that **the self-consistency level achieved by integrative decoding is significantly better than the other approaches** on six LLMs. Notably, the second best approach in terms of self-consistency level is SE-RG, but demands significantly more inference costs than ID. According to our efficiency analysis, ID is able to achieve superior self-consistency while consuming only 13.72% of the inference latency required by SE-RG.
>
> [9] SelfCheckGPT: Zero-resource black-box hallucination detection for generative large language models.
>
> [10] Detecting hallucinations in large language models using semantic entropy.

---

> > ### Author Response · Authors · 2024-11-21
> > **Case Study of USC and SR**
> >
> > The outputs of USC and SR in the case shown in Table 8 are as follows. We also list more examples generated by different approaches in Appendix F.
> >
> > **USC**: David Parnas introduced the concept of “information hiding” in software engineering, emphasizing the importance of encapsulating internal details of a system from its users **(✓)**. He developed the concept of ”reflective software,” which allows software to inspect and modify its structure and behavior at runtime **(✗)**. Parnas was a pioneer in the field of software architecture, advocating for the importance of understanding and managing the structure and organization of complex software systems. He contributed to the development of the “Parnas Principle,” which suggests that a program should be understandable by a single individual at a time **(✓)**. Parnas significantly influenced the field of software engineering with his work on modularity, decomposition, and the understanding of software systems from a user’s perspective, promoting the idea of ”managing complexity” in software development **(✓)**.
> >
> > **SR**: David Parnas introduced the concept of modularity, emphasizing that programs should be structured into manageable, interchangeable components **(✓)**. He proposed the principle of information hiding, suggesting that the implementation details of a module should not be visible to other modules, enhancing the system's maintainability and security **(✓)**. Parnas developed the idea of levels of abstraction, which is crucial for understanding complex systems by breaking them down into comprehensible layers **(✓)**. He contributed to the development of data flow analysis, a technique used in compilers and software engineering to analyze the flow of data through a program **(✗)**. This principle, advocated by Parnas, suggests that software should behave in a predictable manner, minimizing unexpected behavior, which enhances user satisfaction and system reliability **(✓)**.

---

> ### Author Response · Authors · 2024-11-21
> **Exaplanation of “Context Length” in Table 1**
>
> Thanks for Reviewer Q7YL’s question.
>
> In Table 1, "context length" is meant to compare the context length requirement of a particular method with the one of a vanilla prompting approach.  For vanilla prompting, the required context length is the sum of the prompt length and the response length. Integrative decoding extends this by the length of an additional sampled response, making it approximately twice that of the standard prompting scenario. However, for USC and SR, the context length needs to be extended by k times the sampled response length. Thus, they should be actually marked as “×(k+1)”.  Our intention is to highlight that integrative decoding demands much less of the long-text processing capability of the LLM.
>
> We will definitely include clearer illustration regarding this point in our revision.

---

> ### Author Response · Authors · 2024-11-24
>
> Thank you, Reviewer Q7YL, for providing us with further feedback!
>
> We have revised our manuscript and incorporated new results and explanations based on your suggestions:
>
> - **Inference Efficiency**: We include the analysis of inference cost is in Section 3.4, and further discuss the value of exploring techniques to utilize more inference-time computation in exchange of enhanced performance in Appendix D.1.
> - **Evaluation of Language Coherence:** The evaluation results of language coherence are detailed in Table 4.
> - **Additional Experiments Supporting the Decoding Objective of ID**: We demonstrate that ID can effectively foster both self-consistency and language coherence in its decoding objective through comprehensive experiments in Section 3.6.
> - **Other Revisions**: We add the outputs of SR and USC in the case study of Table 12. We revise the confusing expression of “context length” into “input length” in Table 1.
> - **Additional Baselines**: In addition, we have conducted more experiments to compare ID with several very recent techniques for improving factuality, as shown in Table 2, which also present very promising results.
>
> We hope these revisions have adequately addressed your concerns about our work. If you have any further queries, we would be delighted to engage in continued discussions.
>
> We would greatly appreciate it if you could reconsider the rating of our work, taking our new revisions into account. Nonetheless, in either cases, we are sincerely grateful for your professional advice, which has significantly helped us make our work more solid and comprehensive. Thanks again for your invaluable time and effort throughout the review process.
>
> Sincerely,
>
> Authors of Paper #2558

---

> ### Comment · Reviewer_Q7YL · 2024-12-03
>
> I have raised the score.

---

### Official Review · Reviewer_kiKS · 2024-10-21

**Soundness:** 3
**Presentation:** 3
**Contribution:** 3
**Rating:** 6
**Confidence:** 4

**Summary:**

This work proposes integrative decoding (ID), an extension of leveraging self-consistency to enhance the factuality of LLM generations. Specifically, the authors first sample N responses and subsequently use each as a prefix for prompting the LLM to answer the same question again. Finally, a better response is created by ensembling the generations at token-level. Experimental results on three benchmarks validate the effectiveness of the proposed method.

**Strengths:**

1. The writing is clear, and the proposed method is simple and effective.
2. Rather than selecting a response from a set of candidates, ID integrates them through generation, which is interesting.

**Weaknesses:**

1. The main limitation is the lack of comparison with atomic self-consistency approaches [1][2][3]. The study only compares its method with some older baselines (e.g., USC and SR), which have been significantly surpassed by more recent works.
2. The related work section could benefit from discussing the use of self-consistency for uncertainty estimation (e.g., [4]). Self-consistency is not only an approach to improve factuality but is also considered a better estimator of LLM confidence. Incorporating these discussions could potentially deepen the motivation behind this research direction.


[1] Atomic Self-Consistency for Better Long Form Generations.
[2] Integrate the Essence and Eliminate the Dross: Fine-Grained Self-Consistency for Free-Form Language Generation.
[3] Improving LLM Generations via Fine-Grained Self-Endorsement.
[4] LUQ: Long-text Uncertainty Quantification for LLMs

**Questions:**

Please see the above-mentioned weakness.

---

> ### Author Response · Authors · 2024-11-19
> **Comparison with Additional Baselines (1/2)**
>
> We would like to thank Reviewer kiKS for directing us to more recent related research, allowing us to make our work more comprehensive. First, we would like to address the reviewer's primary concern about our compared baselines by providing some additional experimental results.
>
> ## Additional Baselines
>
> We further compare our proposed approach (**ID**) with the following baselines:
>
> - **SE-SL** [1]: This approach first prompts the LLM to divide the response into a sequence of facts and then calculates a self-endorsement score for each response, by checking the consistency between each fact within it and all other sampled responses through LLM prompting. The response with the highest self-endorsement score is selected as the final output.
> - **SE-RG** [1]: It is a variant of SE-SL. Instead of selecting one of the sampled response as the final output, it regenerates a new output with some of the selected facts.
> - **FSC** [2]: It instructs the LLM to extract common segments among sampled responses and regenerate a new output accordingly.
>
> For SE-SL and SE-RG, we follow the original implementation [1] by setting the number of sampled responses to 10 and the threshold for the self-endorsement score used to select reference facts to 0.8. For FSC, we set the number of sampled responses with the optimal one on the validation set by selecting from {4, 5, 8, 12, 16} on TruthfulQA and Biographies; directly set it five on the Longfact, following their original implementation [2], due to the high inference cost on Longfact.
>
> We regret that we are unable to include **ASC** [3], another recent approach mentioned by Reviewer kiKS, in our new experiments, as this work was only recently published at EMNLP 2024 in November, and the implementation code has not yet been released (the GitHub repository provided in the paper is still empty).
>
> [1] Improving LLM Generations via Fine-Grained Self-Endorsement.
>
> [2] Integrate the Essence and Eliminate the Dross: Fine-Grained Self-Consistency for Free-Form Language Generation.
>
> [3] Atomic Self-Consistency for Better Long Form Generations.
>
> ## Evaluation Results
>
> We report the evaluation results in the following table.
>
> |  |   | /-------   | TruthfulQA |  -------/ | /--   Biogra | phies --/ | /--------   | Longfact |  --------/ |
> |  :---- |  :----   |  :----:  | :----: | :----:  | :----: |:----: | :----:  | :----: | :----:  |
> | **Backbone** | **Method**  |  **%Truth**   | **%Info** | **%T * I**  | **# Correct** | **% Acc** |  **Prec.**  |  **R@128**   | **F1@128** |
> | **Llama2** |  + SE-SL   | 50.5 | 96.1 | 48.5 |0.75 | 15.0 | 88.2 | 74.7 | 81.1 |
> | |   + SE-RG   | 45.4 | 94.6 | 42.9 |0.82 | 16.4 | 85.2 | 54.5 | 64.8 |
> | |  + FSC  | 52.4 | 95.6 | 50.1 |0.82 | 16.4 | 88.0 | 64.0 | 72.6 |
> | |  + ID | **55.9** | **99.0** | **55.3** | **0.87** | **17.3** | **89.0** | **77.5** | **82.1** |
> |  |     |    |  |  |
> | **Llama3** |  + SE-SL  | 58.0 | 98.3 | 57.1  | 1.48 | 32.8 | 92.5 | 68.0 | 77.7 |
> | |   + SE-RG   | 54.4 | 96.3 | 52.4 |1.60 | 34.5 |91.8 | 47.7 | 62.0 |
> | |  + FSC  | 56.5 | 93.4 | 52.8 | 1.33 | 27.9 |  **92.5** | 47.3 | 60.2 |
> | |  + ID | **63.4** | **99.0** | **62.8** | **2.00** | **42.0** | 92.2 | **77.7** | **83.6** |
> |  |     |    |  |  |
> | **Gemma2** |  + SE-SL  | 69.8  | 98.3 | 68.3  |2.29 | 47.3 |  97.1 | 56.1 | 70.3 |
> | |   + SE-RG   | 70.5  | 97.8 | 68.9  |2.40 | 50.5 | 96.7 | 42.6 | 58.4 |
> | |  + FSC  | 69.8  | 98.3 | 68.3  |1.70 | 36.0 | 95.8 | 50.4 |  65.1 |
> | |  + ID   | **77.1**  | **99.0** | **76.3**  | **2.52** | **52.4** |  **97.1** | **69.7** | **80.4** |
> |  |     |    |  |  |
> | **GLM4** |  + SE-SL  | 61.0 | 98.5 | 60.1 | 1.37 | 27.3 | 88.9 | 62.5 | 72.9 |
> | |   + SE-RG   | 64.1 | 97.8 | 62.7 | 1.36 | 27.2 | 88.0 | 48.7 | 62.1 |
> | |  + FSC   | 63.4 | 97.8 | 62.0 |   1.58 | 31.7 |  **90.3** | 38.4 | 52.8 |
> | |  + ID | **65.1** | **99.0** | **64.5** |  **1.81** | **36.2** | 89.2 | **66.4** | **75.9** |
> |  |     |    |  |  |
> | **Mistral** |  + SE-SL  | 76.8 | 99.5 | 76.8 |**1.16** | **23.3** | 91.6 | 58.5 | 70.6 |
> | |   + SE-RG  | 72.9 | 97.8 | 71.3 | 1.10 | 22.0 | 90.9 | 44.2 | 58.6 |
> | |  + FSC  | 78.0 | 99.5 | 77.7 | 0.87 | 17.5 | 91.3 | 57.8 | 69.1 |
> | |  + ID  | **78.8** | **99.5** | **78.4** |  1.11 | 22.6 | **91.8** | **68.5** | **77.7** |
> |  |     |    |  |  |
> | **Qwen2** |  + SE-SL  | 57.1 | 97.1 | 55.4 |  1.48 | 29.5 | 91.2 | 55.9 | 68.2 |
> | |   + SE-RG  | **62.9** | 94.9 | **59.7** | 1.54 | 30.8 | 91.3 | 44.3 | 57.9 |
> | |  + FSC  | 57.3 | 98.0 | 56.2 |  1.55 | 31.1 | 91.3 | 38.6 | 52.0 |
> | |  + ID | 60.0 | **99.0** | 59.4 | **1.74** | **35.5** | **91.7** | **64.2** | **74.8** |
> |  |     |    |  |  |

---

> > ### Comment · Reviewer_kiKS · 2024-11-23
> >
> > I have raised the score. Please also update these results in the manuscript. By the way, I still suggest including the results of ASC. The paper is public on arXiv (2405.13131) and seems not hard to implement.

---

> ### Author Response · Authors · 2024-11-19
> **Comparison with Additional Baselines (2/2)**
>
> Based on the results shown in the table above, we highlight some key experimental findings:
>
> - **Integrative decoding (ID) still demonstrates the best overall performance in terms of factual accuracy**. It is only in very few cases (i.e. Mistral on Biographies and Qwen2 on TruthfulQA) that ID is surpassed by SE-SL or SE-RG, but the performance still remains comparable. In addition, we want to emphasize that **ID is much more efficient and simpler to implement than SE-SL and SE-RG**, as these two methods require numerous iterations of LLM inference to break down each sampled response into a series of facts and verify the consistency among them. In contrast, ID does not rely on additional prompting to check self-consistency, making it applicable to a wider range of scenarios.
> - **The advantages of ID are particularly notable in document-level generation tasks.**  Enhancing factuality on long-form generation tasks is very challenging and less explored. As evidenced by the table above, previous methods have struggled with the LongFact benchmark, which requires comprehensive document-level responses. **Although previous approaches can also enhance factual precision, they often result in a marked decline in information recall**. This indicate that they need to sacrifice a large degree of informativeness to ensure factual accuracy. In contrast, ID maintains a robust balance between factual accuracy and informativeness, leading to improvements in both dimensions.

---

> ### Author Response · Authors · 2024-11-19
> **Additional Related Works**
>
> We appreciate Reviewer kiKS’s suggestion to discuss the role of self-consistency in uncertainty estimation in the realted work section, which would enhance the motivation behind our work.
>
> In our current manuscript, we have already mentioned several works that utilize self-consistency as an indicator of factuality (lines 107–116) [4][5]. These works form the foundation and preliminaries of our proposed method. However, we agree that expanding on this research direction in the related work section would enhance its comprehensiveness.
>
> Apart from [6] mentioned by Reviewer kiKS, we would also include more related studies, such as [7-13].
>
> [4] SelfCheckGPT: Zero-resource black-box hallucination detection for generative large language models
>
> [5] Detecting hallucinations in large language models using semantic entropy
>
> [6] LUQ: Long-text Uncertainty Quantification for LLMs
>
> [7] Semantic uncertainty: Linguistic invariances for uncertainty estimation in natural language generation.
>
> [8] Generating with confidence: Uncertainty quantification for black-box large language models
>
> [9] Self-contradictory hallucinations of large language models: Evaluation, detection and mitigation
>
> [10] Shifting attention to relevance: Towards the predictive uncertainty quantification of free-form large language models
>
> [11] How can we know when language models know? on the calibration of language models for question answering
>
> [12] Calibration of pre-trained transformers
>
> [13] Uncertainty-aware machine translation evaluation

---

> ### Author Response · Authors · 2024-11-24
>
> Thank you, Reviewer kiKS, for your feedback.
>
> We have revised our manuscript and updated Table 2 with the new results. Additionally, we are committed to reproducing the implementation of ASC. We appreciate your invaluable suggestions, which have helped us make our work more comprehensive.
>
> Sincerely,
>
> Authors of Paper #2558

---

### Official Review · Reviewer_TZMW · 2024-11-03

**Soundness:** 2
**Presentation:** 3
**Contribution:** 2
**Rating:** 6
**Confidence:** 3

**Summary:**

In this work, the authors propose a new decoding method called "Integrative Decoding" so that self-consistency can be obtained. The experiments shows that this decoding method can improve factuality. The improvements are shown in three benchmarks (TruthfulQA, Biographies, and LongFact).

**Strengths:**

1. A new decoding method is proposed to improve consistency. Higher factuality are obtained in the experiments.

2. The evaluations are done over six series of LLM with varying scales and multiple benchmarks (TruthfulQA, Biographies, and LongFact).

3. In the analysis, as the number of sampled responses increases, the performance is improved.

**Weaknesses:**

1. It is reasonable that consistency is obtained with this decoding method. However, factuality improvement seems unclear. Improvements seems more possible If the model is well calibrated. Some discussions can be done in this perspective.

2. In the evaluation of these experiments, LLM (GPT-4) is used for factuality evaluation. However, there is no human annotation results. Miss match could happen for the evaluation. It could cause the actual improvements.

3. The cost of the proposed decoding method is high: double context length is need, and more inference steps (the same as the number of sampled responses). For long-form generation, it is hard and inefficient.

**Questions:**

According to Figure 3, the claim "The performance of integrative decoding progressively improves with more sampled responses across six LLMs." seems wrong. For the three models (Llama3, Gemma2, and GLM4), accuracy is increasing, then decreasing. Could you have a check?

---

> ### Author Response · Authors · 2024-11-20
> **Factuality Improvement and Evaluation Approaches**
>
> We appreciate Reviewer TZMW’s advice on further discussing the reliability of our evaluation approach. We will definitely include this in our next revision. For now, we would like to address this concern as follows:
>
> Firstly, we want to clarify GPT-4's role in our evaluation process.  Rather than relying on GPT-4's intrinsic parametric knowledge, we provide it with **sufficient reference information necessary for assessment** to conduct evaluation. In other words, it only needs to check whether the assessed content is supported by the reference. As illustrated in Appendix B, on TruthfulQA, we included the reference correct answers and typical wrong answers annotated in the dataset as reference, guiding GPT-4 in its evaluation. On Biographies, where the model is required to generate five major achievements of a particular scientist, GPT-4 evaluates the factuality by referring to the information extracted from Wikipedia.
>
> Evaluating factuality in free-form text generation is inherently challenging and resource-intensive. **Leveraging powerful LLMs like GPT-4, as we did, to evaluate factuality with reference information is a well-established and widely-accepted evaluation standard within the community.** Current language models are sufficiently capable of performing tasks like accuracy verification according to reference material. Numerous studies have adopted similar automated evaluation standards, such as those found in [1-5]. **Therefore, we uphold the reliability of our approach, and the improvement in factuality achieved is evident**.
>
> Furthermore, we conducted extensive experiments across various LLMs and compared our method with other hallucination mitigation strategies using the same evaluation criteria. Given the consistent and significant performance improvements exhibited by our approach in such a great number of experiments, it is highly unlikely that these advantages are soley due to discrepancies in automatic evaluation.
>
> In response to Reviewer TZMW’s suggestion, we will also add a set of human evaluation results in our next revision. However, since human evaluation, especially for free-form generation, is exceptionally time-consuming, this addition may require additional time to complete. We appreciate your patience and understanding as we work towards this enhancement.
>
> [1] Improving factuality and reasoning in language models through multiagent debate
>
> [2] Improving LLM Generations via Fine-Grained Self-Endorsement.
>
> [3] LUQ: Long-text Uncertainty Quantification for LLMs
>
> [4] TruthfulQA: Measuring How Models Mimic Human Falsehoods
>
> [5] DoLa: Decoding by Contrasting Layers Improves Factuality in Large Language Models

---

> ### Author Response · Authors · 2024-11-20
> **Inference Cost and Efficiency Issue (1/2)**
>
> To address Reviewer TZMW’s concern on the inference cost of our approach, our response is as follows.
>
> First of all, we want to underscore that **being able to trade off inference cost for performance enhancement should not be indiscriminately considered as a weakness** of a techinque. This is particularly true when its inference cost is comparable to other existing improvement methods and the performance gains are substantial.
>
> In fact, exploring ways to **utilize more inference-time computation in exchange of enhanced performance is a promising and rapidly growing research direction** [6-8]**,** as demonstrated by the recent success of o1 [9]. The potential of these approaches extends beyond merely pushing the performance boundaries of existing language models. More importantly, they offer practitioners new perspectives and greater flexibility when balancing inference cost and performance.  For instance, as shown in Figure 5 of our paper, our approach can enhance the performance of Llama2-13B more effectively than the much larger model Llama2-70B. Meanwhile, the inference cost of applying our method to Llama2-13B can be even lower than conducting a single inference iteration on Llama2-70B in many scenarios.
>
> To further address Reviewer TZMW’s concern on the inference cost of our approach, we have undertaken additional experiments to assess its efficiency and compare it with other methods that leverage self-consistency to enhance factuality. In addition to the two baseline methods included in the current manuscript (i.e., **USC** [10] and **SR** [11]), three additional recent approaches have been incorporated for comparative analysis, as per the recommendation of Reviewer kiKS.
>
> - **SE-SL** [12]: This approach first prompts the LLM to divide the response into a sequence of facts and then calculates a self-endorsement score for each response, by checking the consistency between each fact within it and all other sampled responses through LLM prompting.
> - **SE-RG** [12]: It is a variant of SE-SL. Instead of selecting one of the sampled response as the final output, it regenerates a new output with some of the selected facts.
> - **FSC** [13]: It instructs the LLM to extract common segments among sampled responses and regenerate a new output accordingly.
>
> We apply these approaches on Llama3 to perform inference on the TruthfulQA benchmark, using a single GPU of A100 80GB. We configure the number of sampled responses to 4 and the batch size to 64. The following metrics are taken into consideration for analysis:
>
> - **Lantency/sample (ms)**: measures the average inference time for each sample.
> - **Latency/token (ms)**: gauges the average inference time for each token generated in the final output. Tokens generated in intermediate steps and chain-of-thought reasoning excluded to ensure a fair comparison.
> - **Throughput (token/s)**: calculates the average number of tokens generated per second.
> - **Factuality Improvement** (%): represents the absolute improvement in the T*I metric on the TruthfulQA benchmark. We list the factuality improvement yielded by each method for reference to analyze the trade-off between inference cost and performance enhancement.
>
> The results are shown in the following table:
>
> | Method | Latency/sample (ms) ↓ | Latency/token (ms) ↓ | Throughput (token/s) ↑ | Factuality Improvement (%) ↑ |
> |--------|-----------------------|---------------------| ---------------------- |------------------------------|
> | Greedy | 5.815 (×1.00)                | 0.102 (×1.00)               | 975.76 (×1.00)                 | -                            |
> | USC    | **58.35 (×10.03)**        | **0.928 (×9.10)**       | **107.73 (×0.11)**         | 4.3                          |
> | SR     | 96.96 (×16.67)        | 1.965 (×19.26)      | 50.90 (×0.05)          | 4.5                          |
> | FSC    | 96.98 (×16.68)        | 1.965 (×19.26)      | 50.88 (×0.05)          | 0.8                          |
> | SE-SL  | 493.17 (×84.81)       | 8.374 (×82.09)      | 11.96 (×0.01)          | 5.5                          |
> | SE-RG  | 510.98 (×87.87)       | 7.277 (×71.35)      | 13.74 (×0.01)          | 1.2                          |
> | **ID**     | **68.26 (×11.74)**       | **1.127 (×11.04)**      | **86.78 (×0.09)**          | **11.2**                         |
>
> We can see that the inference cost of our method (ID) is comparable to USC and significantly lower than all other methods that utilize self-consistency for hallucination mitigation. It is because those methods necessitate numerous additional iterations of inference or extensive chain-of-thought reasoning to assess consistency among sampled responses, while our method does not. Furthermore, the enhancement in factual accuracy achieved by ID greatly surpasses that of the other methods. These results demonstrate that ID effectively balance efficiency and performance enhancement, compared with other approaches in this line of research.

---

> > ### Author Response · Authors · 2024-11-20
> > **Inference Cost and Efficiency Issue (2/2)**
> >
> > [6] Scaling LLM Test-Time Compute Optimally can be More Effective than Scaling Model Parameters
> >
> > [7] Large Language Monkeys: Scaling Inference Compute with Repeated Sampling
> >
> > [8] Are More LLM Calls All You Need? Towards Scaling Laws of Compound Inference Systems
> >
> > [9] Learning to Reason with LLMs
> >
> > [10] Universal self-consistency for large language model generation.
> >
> > [11] Self-refine: Iterative refinement with self-feedback.
> >
> > [12] Improving LLM Generations via Fine-Grained Self-Endorsement.
> >
> > [13] Integrate the Essence and Eliminate the Dross: Fine-Grained Self-Consistency for Free-Form Language Generation.

---

> ### Author Response · Authors · 2024-11-22
>
> Thank you for pointing out the issue with our claim regarding performance improvement with repeated sampling. We will revise the statement to be more precise and approprate. In fact, the performance improvement may plataus after reaching a certain number of sampled responses and then flutuate at this level with additional sampling. Our intended message is that there is a general trend of performance improvement with increased sampling. More importantly, we want to emphasize that, **compared with the baseline methods, the improvement trend resulting from our approach is more significant and stable**.

---

> ### Author Response · Authors · 2024-11-25
> **Additional Human Evaluation Results**
>
> In response to Reviewer TZMW's concern on the reliability of automatic evaluation, we further performed human evaluation on the TruthfulQA dataset for ID and five strong baseline approaches: USC, SR, FSC, SE-SL, and SE-RG. We used LLaMA3-8B as the base model and included 128 samples from the TruthfulQA test set in our evaluation. We recruited three undergraduate computer science students, who were not involved in our research project, to carry out the evaluation. They were provided with the reference correct answers and the typical wrong answers for each question to aid in their assessment process. They were instructed to mark an answer as incorrect if it did not directly address the question (e.g., “I’m sorry. I don’t know”). The inter-annotator agreement achieved a Fleiss’ Kappa score of 0.769, indicating strong agreement.
>
> The evaluation results are presented in the following table.
> | Method | Truthful (%) |
> |--------|--------------|
> | USC    | 59.38        |
> | SR     | 64.06        |
> | SE-SL  | 60.94        |
> | SE-RG  | 55.47        |
> | FSC    | 60.16        |
> | **ID**     | **65.62**        |
>
> The human evaluation also indicates that the performance of ID is significantly better than the other approaches.
>
> Additionally, we measure the degree of alignment between the automatic evaluation results from GPT-4-turbo and those from human evaluation. We observed that the matching rates between them range from 90.62\% to 94.53\%. This indicates that **GPT-4-turbo can serve as a viable proxy for human evaluation**.

---

> > ### Comment · Reviewer_TZMW · 2024-11-27
> >
> > Thanks for the added human evaluation results. It solves my concern about automatic evaluation results.

---

> ### Author Response · Authors · 2024-11-25
> **Kindly Request Feedback from Reviewer TZMW**
>
> Dear Reviewer TZMW,
>
> We sincerely appreciate the time and effort you have dedicated to reviewing our manuscript. We have now submitted a revised version of our paper, which includes the following updates based on your insightful feedback:
> - **Inference Efficiency**: We include the analysis of inference cost is in Section 3.4, and further discuss the value of exploring techniques to utilize more inference-time computation in exchange of enhanced performance in Appendix D.1.
> - **Additional Human Evaluation and Discussion on the Evaluation Reliability**: We include human evaluation results in Appendix C.1 and discuss the evaluation reliability in Appendix D.2.
> - **Other Revisions**: We revise our claim regarding performance improvement with repeated sampling to be more claim in lines 375~376.
>
> We hope that these revisions address your concerns effectively. We kindly request that you reconsider the rating of our work in light of these updates.
>
> Thank you for your considering our response in the rebuttal period.
>
> Sincerely,
>
> Authors of Paper #2558

---

> ### Author Response · Authors · 2024-11-26
> **Kindly Request Feedback from Reviewer TZMW**
>
> Dear Reviewer TZMW,
>
> We sincerely appreciate the time and effort you have dedicated to reviewing our manuscript. We have worked hard in the past two weeks to add exeperiments and respond to your concerns.
>
> We kindly request that you might give us any possible feedback and reconsider the rating of our work in light of our efforts in the rebuttal period.
>
> Sincerely,
>
> Authors of Paper #2558

---

> ### Author Response · Authors · 2024-11-28
>
> Dear Reviewer TZMW,
>
> Thank you for providing us with additional feedback. **We are glad that you could acknowledge that our efforts in the rebuttal period have effectively addressed your concerns on our evaluation results**. In light of this, we believe that we have addressed the primary issues highlighted in your previous comment (Weaknesses 1 and 2).
>
> However, we are disheartened to note that **the rating you provided for our work is still below the acceptance threshold (rating: 5, contribution: 2, soundness: 2)**.
>
> We would be most grateful if you could kindly share any insights or reasons that influenced this decision. Your guidance would be invaluable to us as we strive to enhance our work further. Thank you.
>
> Sincerely,
>
> Authors of Paper #2558

---

> ### Author Response · Authors · 2024-12-01
>
> Dear Reviewer TZMW,
>
> Thank you for providing us with additional feedback. **We are glad that you could acknowledge that our efforts in the rebuttal period have effectively addressed your concerns on our evaluation results**. In light of this, we believe that we have addressed the primary issues highlighted in your previous comment (Weaknesses 1 and 2).
>
> However, we are disheartened to note that **the rating you provided for our work is still below the acceptance threshold (rating: 5, contribution: 2, soundness: 2)**.
>
> We would be most grateful if you could kindly share any insights or reasons that influenced this decision. Your guidance would be invaluable to us as we strive to enhance our work further. Thank you.
>
> Sincerely,
>
> Authors of Paper #2558

---

> > ### Comment · Reviewer_TZMW · 2024-12-02
> >
> > I just updated the review. Thanks for the active response.

---

### Official Review · Reviewer_JqQZ · 2024-11-03

**Soundness:** 4
**Presentation:** 3
**Contribution:** 3
**Rating:** 8
**Confidence:** 4

**Summary:**

The "Self-consistency" approach consists in  sampling from a LLM  a set of many different outputs  to select the final hypothesis as the most consistent one with respect to this set. This kind of approach  really helps to improve the "factuality"  of the output.
This paper builds on this line of research to Integrative Decoding (ID). It allows the use of self consistency approach text generation tasks. The idea :-) is to extend the decoding objective with a consistency term. This new objective can be estimated with a smart and integrative prompt. Experiments show nice  improvements on many benchmarks like TruthfulQA, Biographies and LongFact.

**Strengths:**

This idea is simple and effective. The theoretical formulation is clear, while in practice the solution is relatively straightforward.  The experimental results clearly support the claims.

**Weaknesses:**

The extra cost of this approach is maybe not worst than the other self-consistency based methods. However this drawback could be discussed in the paper. It could be nice to provide inference time, even if it is not the claim of the paper.

**Questions:**

In equation 4, there are two terms f and G. Then their sum is approximated by the inference with the new prompt. Did you try to recover these two terms given this approximation, to see if your motivation sounds ?

---

> ### Author Response · Authors · 2024-11-19
> **Analysis of Inference Cost**
>
> We would like to express our gratitude to Reviewer JqQZ for recognizing our work and contribution. Additionally, we value Reviewer JqQZ's recommendation regarding the analysis of the inference cost associated with our methodology. In light of this feedback, we have undertaken additional experiments to assess the efficiency of our approach and compare it with previous works that leverage self-consistency to enhance factuality.
>
> In addition to the two baseline methods included in the current manuscript (i.e., **USC** [1] and **SR** [2]), three additional recent approaches have been incorporated for comparative analysis, following Reviewer kiKS's suggestion:
> - **SE-SL** [3]: This approach first prompts the LLM to divide the response into a sequence of facts and then calculates a self-endorsement score for each response, by checking the consistency between each fact within it and all other sampled responses through LLM prompting. The response with the highest self-endorsement score is selected as the final output.
> - **SE-RG** [3]: It is a variant of SE-SL. Instead of selecting one of the sampled response as the final output, it regenerates a new output with some of the selected facts.
> - **FSC** [4]: It instructs the LLM to extract common segments among sampled responses and regenerate a new output accordingly.
>
> We apply these approaches on Llama3 to perform inference on the TruthfulQA benchmark, using a single GPU of A100 80GB. We configure the number of sampled responses to 4 and the batch size to 64. The following metrics are taken into consideration for analysis:
>
> - **Lantency/sample (ms)**: measures the average inference time for each sample.
> - **Latency/token (ms)**: gauges the average inference time for each token generated in the final output. Only tokens within the final produced answer are considered. Tokens generated in intermediate steps and chain-of-thought reasoning excluded to ensure a fair comparison.
> - **Throughput (token/s)**: calculates the average number of tokens generated per second.
> - **Factuality Improvement (%)**: represents the absolute improvement in the T*I metric on the TruthfulQA benchmark. We list the factuality improvement yielded by each method for reference to analyze the trade-off between inference cost and performance enhancement.
>
> The results are shown in the following table
>
> | Method | Latency/sample (ms) ↓ | Latency/token (ms) ↓ | Throughput (token/s) ↑ | Factuality Improvement (%) ↑ |
> |--------|-----------------------|---------------------| ---------------------- |------------------------------|
> | Greedy | 5.815 (×1.00)                | 0.102 (×1.00)               | 975.76 (×1.00)                 | -                            |
> | USC    | **58.35 (×10.03)**        | **0.928 (×9.10)**       | **107.73 (×0.11)**         | 4.3                          |
> | SR     | 96.96 (×16.67)        | 1.965 (×19.26)      | 50.90 (×0.05)          | 4.5                          |
> | FSC    | 96.98 (×16.68)        | 1.965 (×19.26)      | 50.88 (×0.05)          | 0.8                          |
> | SE-SL  | 493.17 (×84.81)       | 8.374 (×82.09)      | 11.96 (×0.01)          | 5.5                          |
> | SE-RG  | 510.98 (×87.87)       | 7.277 (×71.35)      | 13.74 (×0.01)          | 1.2                          |
> | **ID**     | **68.26 (×11.74)**       | **1.127 (×11.04)**      | **86.78 (×0.09)**          | **11.2**                         |
>
> We can see that **the inference cost of our method (ID) is comparable to USC and significantly lower than all other methods that utilize self-consistency for hallucination mitigation**. It is because those methods necessitate numerous additional iterations of inference or extensive chain-of-thought reasoning to assess consistency among sampled responses, while our method does not. Furthermore, the enhancement in **factual accuracy achieved by ID greatly surpasses that of the other methods**. These results demonstrate that ID effectively balance efficiency and performance enhancement, compared with other approaches in this line of research.
>
> [1] Universal self-consistency for large language model generation.
>
> [2] Self-refine: Iterative refinement with self-feedback.
>
> [3] Improving LLM Generations via Fine-Grained Self-Endorsement.
>
> [4] Integrate the Essence and Eliminate the Dross: Fine-Grained Self-Consistency for Free-Form Language Generation.

---

> ### Author Response · Authors · 2024-11-21
> **Additional Experiments to Confirm the Decoding Objective of ID (1/2)**
>
> In response to Reviewer JqQZ’s question, we did additional experiments to confirm confirm integrative decoding is indeed able to encourage both **language coherence** and **self-consistency** at the same time. To this end, we conduct two additional sets of experiments to assess these two dimensions, respectively.
>
> ### Evaluation of Language Coherence
>
> We compare the outputs generated through integrative decoding and greedy decoding for each sample in the test set of TruthfulQA in terms of language fluency and coherence, using GPT-4-turbo. Specifically the template we employ to prompt GPT-4 for evaluation is as follows:
>
> > Text A: {text_a}
> >
> > Text B: {text_b}
> >
> > Which of the two texts is more coherent and fluent in terms of language use, Text A or Text B? Focus solely on language use. You do not need to consider the factual accuracy of the text. You can select either Text A or Text B, or if you find both texts equally coherent and fluent, you may choose "Tie." However, you are encouraged to select one of the two texts.
> >
> > Your answer should be either "A", "B", or "Tie". After choosing, briefly explain your decision. Then you can explain your choice with a few words.`
>
> Note that the outputs from integrative decoding and greedy decoding are randomly assigned to the positions of {text_a} and {text_b} to eliminate position bias.
>
> The evaluation results are shown in the following table.
>
> | Base Model | ID wins | Tie | Greedy wins |
> | --- | --- | --- | --- |
> | Gemma2 | 11.95 | 80.49 | 7.56 |
> | GLM4 | 16.34 | 72.68 | 10.98 |
> | Llama2 | 12.68 | 82.44 | 4.88 |
> | Llama3 | 8.54 | 82.93 | 8.54 |
> | Mistral2 | 11.22 | 76.83 | 11.95 |
> | Qwen2 | 14.39 | 74.63 | 10.98 |
>
> We observe that most comparisons result in a "Tie," and the number of instances where ID is victorious is slightly higher than those where it isn't. This indicates that the generations from **integrative decoding can achieve the same level of language fluency and coherence as greedy decoding**. We have uploaded all the outputs from both ID and greedy decoding, along with the evaluation results, to the supplementary materials under the directory of “eval_coherence/”.

---

> ### Author Response · Authors · 2024-11-21
> **Additional Experiments to Confirm the Decoding Objective of ID (2/2)**
>
> ### Evaluation of Self-consistency
>
> We measure the degree of self-consistency between an assessed output and a set of sampled responses, following [5, 6]. Formally, given a set of sampled responses $\mathcal{R} = \{r_1, r_2, ..., r_k\}$ and an output $y$ that encompass a set of facts $y=\{s_1, s_2,...,s_n\}$, we define the self-consistency score of $y$ as:
>
> $$
> SC(y,\mathcal{R})=\frac{1}{k\cdot n}\sum_{i=1}^n\sum_{j=1}^k \text{consistency}(s_i,r_j),
> $$
>
> where SC($\cdot$) represents the self-consistency score. $\text{consistency}(s_i,r_j)$ denotes whether $y$ is supported by $r_j$. It return 1 as 1 if $s_i$ is supported by $r_j$, 0 if $y$ contradicts  $r_j$, and 0.5 if the relationship is inconclusive. We employ GPT-4-turbo to assess $\text{consistency}(s_i,r_j)$ through the following prompt template:
>
> > Take the following facts about a person as truth: {premise}.
> >
> > Please check the consistency between the text above and the fact "{hypothesis}".
> >
> > Choose one of the following answers:
> >
> > A. The fact is supported by the text above.
> >
> > B. The fact is contradicted by the text above.
> >
> > C. The fact is neither supported nor contradicted by the text above. It is inconclusive.
> >
> > Your answer should be one word ("A", "B" or "C").`
>
> We conduct evaluation on **ID** and the baseline approaches that aim to enhance self-consistency in the final output (i.e., **USC**, **SR**, **SE-SL**, **SE-RG**, **FSC**). The evaluation is conducted on the Biographies benchmark, which requires the model to list five major achievement of a scientist. We divide the output $y$ into a set of facts $\{s_1, s_2,...,s_n\}$ by treating each listed major achievement as a separate fact. We consider the scenarios where the factuality improvement approach integrates 8 sampled responses (i.e., $k=8$), and measures the self-consistency between the final output and the eight sampled responses. The sampled responses are obtained through temperature sample, with T=0.7. We also evaluate the self-consistency level between an output that is directly generated through temperature sampling (T=0.7) and the other eight sampled responses, denoted as **Vanilla**.
>
> The evaluation results are as follows:
>
> | Method\Base Model | Llama2 | Llama3 | Mistral | Qwen | Gemma | GLM |
> | --- | --- | --- | --- | --- | --- | --- |
> | Vanilla | 0.6087 | 0.6323 | 0.6024 | 0.6789 | 0.7069 | 0.6453 |
> | USC | 0.6049 | 0.6524 | 0.6064 | 0.6765 | 0.7244 | 0.6641 |
> | SR | 0.6345 | 0.6443 | 0.6509 | 0.7195 | 0.7204 | 0.6948 |
> | FSC | 0.5984 | 0.6343 | 0.6099 | 0.6826 | 0.7097 | 0.6795 |
> | SE-SL | 0.6221 | 0.6715 | 0.6435 | 0.6999 | 0.7481 | 0.6725 |
> | SE-RG | 0.6393 | 0.6466 | 0.6344 | 0.7063 | 0.7520 | 0.6809 |
> | ID | **0.6479** | **0.6821** | **0.6635** | **0.7366** | **0.7592** | **0.7336** |
>
> We can see that **the self-consistency level achieved by integrative decoding is significantly better than the other approaches** on six LLMs. Notably, the second best approach in terms of self-consistency level is SE-RG, but demands significantly more inference costs than ID. According to our efficiency analysis, ID is able to achieve superior self-consistency while consuming only 13.72% of the inference latency required by SE-RG.
>
> [5] SelfCheckGPT: Zero-resource black-box hallucination detection for generative large language models.
>
> [6] Detecting hallucinations in large language models using semantic entropy.

---

### Author Response · Authors · 2024-11-25
**Summary of the Rebuttal Period**

Thank you to all the reviewers for their constructive feedback on our work. Below, we provide a brief overview of: (1) the contributions and strengths of our work as highlighted by the reviewers, and (2) the updates made to the manuscript in response to the reviewers' comments.

## **Summary of Strengths and Contributions**

We propose Integrative Decoding (ID), a decoding algorithm that incorporates "self-consistency" within its decoding objective to enhance the factuality of open-ended generation tasks.

- Our approach is **well-grounded in a clear theoretical framework** while being **simple to implement in practice** (Reviewers JqQZ and kiKS).
- It doesn't need additional prompting or training, making it more **applicable to a wider range of scenarios** than previous works (Reviewer Q7YL).
- We conduct **extensive experiments** over six series of LLM with varying scales and multiple benchmarks (Reviewer TZMW). Our approach **consistently demonstrates significant and stable improvements** (Reviewer Q7YL).


## **Issues Addressed in the Rebuttal Period**

In response to the reviewers' suggestions and concerns regarding our work, we have updated the manuscript mainly in the following aspects:

- **Additional Baselines** (Reviewer kiKS): We further compare ID with three more recently proposed techniques, as shown in ``Table 2``.
- **Additional Experiments to Support the Decoding Objective** (Reviewers JqQZ and Q7YL): We add experiments to support that ID can maintain **language coherence** and effectively foster **self-consistency**, as discussed in ``Section 3.6``.
- **Inference Efficiency** (Reviewers JqQZ, TZMW, and Q7YL): We include an evaluation of inference efficiency in ``Section 3.4``. The results indicate that ID achieves **competitive inference efficiency compared to other techniques in this line of research**, while yielding more significant and stable improvements.  We further discuss the value of exploring techniques to utilize more inference-time computation in exchange of enhanced performance in ``Appendix D.1``.
- **Discussion on the Reliability of GPT-4 Evaluation** and **Human Evaluation** (Reviewer TZMW): We mainly use GPT-4-turbo to asssess factuality, providing it with necessary reference information to guide its evaluation (e.g. reference answers, typical wrong answers, and information extracted from wikipedia). We further discuss the reliability of this evaluation approach in ``Appendix D.2`` and add a set of human evaluation in ``Appendix C.1``.

Thank you once again to all the reviewers for their feedback, which has helped us enhance our work to make it more solid and comprehensive.

---

### Meta-Review · Area_Chair_PPgW · 2024-12-23

**Metareview:**

This paper presents a method for utilizing self consistency via integrative decoding and presents good results on improvements in factuality across several benchmarks.  The paper presents many experiments with comparisons with several baselines.  Overall reviews are positive and I suggest accepting the paper.

**Additional Comments On Reviewer Discussion:**

There is significant back and forth between the authors and the reviewers that led to positive outcome (e.g. raising the score) for the paper, improving it quality.

---

### Decision · Program_Chairs · 2025-01-22

Accept (Poster)